# Sense of control buffers against stress

**Jennifer C Fielder[1]\*, Jinyu Shi[2,3], Daniel McGlade[1,3,4], Quentin JM Huys[5,6†], Nikolaus Steinbeis[1†]**

[1]Division of Psychology and Language Sciences, London, United Kingdom; [2]Department of Experimental Psychology, University of Oxford, Oxford, United Kingdom; [3]Anna Freud, London, United Kingdom; [4]Yale Child Study Center, Yale School of Medicine, Yale University, New Haven, United States; [5]Applied Computational Psychiatry Lab, Mental Health Neuroscience Department, Division of Psychiatry and Max-Planck UCL Centre for Computational Psychiatry and Ageing Research, Queen Square Institute of Neurology, London, United Kingdom; [6]North London NHS Foundation Trust, London, United Kingdom

\*For correspondence:
jennifer.fielder.21@ucl.ac.uk

†These authors contributed equally to this work

## eLife Assessment

This **important** research addresses the effects of subjective control and task difficulty on experienced stress using a novel behavioral task administered on the same day in two large online samples. **Convincing** evidence is provided, establishing the internal and external task validity of the task, as well as a relationship between the sense of control and task difficulty, with individual differences in relevant mental health constructs. Evidence for the specificity of the link between control and stress would be more substantial if the design had not conflated control and reward rate. This work will be of interest to psychologists and clinicians studying the concepts of controllability, stress, and psychopathology.

**Abstract** Stress is one of the most pervasive causes of mental ill health across the lifespan. Subjective dimensions of stress perception, such as perceived control, are especially potent in shaping stress responses. While the impact of reduced or no control over stress is well understood, much less is known about whether heightened feelings of control buffer against the negative impact of later stress. We designed a novel paradigm with excellent psychometric properties to sensitively capture and induce different states of subjective control. Across two studies with a non-clinical sample of 768 adults, we show a robust association between sense of control and stress as well as symptoms of mental ill health. More importantly, in a subsample of 295 participants, we show that compared to a neutral control group, inducing a heightened state of subjective control buffers against the impact of later stress. These findings demonstrate a causal role for a heightened sense of control in mitigating the negative impact of stressful experiences and spell out important directions for future preventative interventions.

## Introduction

Individuals have control when their actions are causally involved in achieving beneficial outcomes or avoiding undesired ones. Having control allows engaging in a suite of proactive goal-directed behaviours and offers rewards for exploration, which the absence of control does not (*Moscarello and Hartley, 2017*; *Teodorescu and Erev, 2014*). Decades of work have demonstrated the motivational significance of experiences of control over one's environment (*Maier and Seligman, 2016*) and their fundamental role in psychological well-being (*de Quadros-Wander et al., 2014*) and resilience (*Yang and Ma, 2020*). The power of control to organise behaviour has been argued to originate from

a fundamental need of humans (and many other species) to be effective in their interactions with the environment (*Elliot and Dweck, 2005*; *Ryan and Deci, 2000*; *Skinner, 1996*). A wealth of cross-species work has also identified having control as a crucial moderator of how potentially stressful events are processed (*Gee et al., 2022*; *Maier and Seligman, 2016*). Despite the significant risk stress poses to mental illness (*Lupien et al., 2009*), the relationship between the experience of stress and control remains poorly understood.

Two core questions remain poorly understood. First, a wealth of evidence suggests that subjective aspects of a range of phenomena, including maltreatment, social hierarchy, loneliness, and pain, appear to be highly predictive of outcomes, such as well-being and mental health, and often more so than objective classifications (*Bzdok and Dunbar, 2022*; *Danese and Widom, 2020*; *Rivenbark et al., 2020*; *Wager and Atlas, 2015*). Indeed, it has long been noted that the impact of objective control will only be relevant if it is perceived and recognised as such (*Langer and Rodin, 1976*). Thus, perceived or subjective dimensions of control play a crucial role in functioning across social, academic, and mental health domains throughout the lifespan (*Bandura et al., 2003*; *Moscarello and Hartley, 2017*; *Rotter, 1966*; *Ryan and Deci, 2006*). However, to date, the affective properties of experiences of control remain poorly understood (although see here for studies on perceptions of control and positive affect *Leotti and Delgado, 2011*; *Ly et al., 2019*; *Wang et al., 2021*; *Wang and Delgado, 2019*). In particular, it is unclear how momentary subjective sense of control relates to subjective reports of stress.

The second question is about the effect of increased control. Converging lines of evidence have shown that subjective control has a critical role in shaping the impact of stressors on a host of different outcomes (*de Quadros-Wander et al., 2014*; *Maier and Seligman, 2016*; *Yang and Ma, 2020*). Systematic experimental work in animals over the last 50 years has also causally demonstrated the critical role of control over stress in determining later outcomes (*Maier and Seligman, 2016*). In classic empirical assays, caged rodents are exposed to shocks that terminate irrespective of any action taken. As a result of such operationalised non-contingency between actions and outcomes, or uncontrollability of a stressor, rodents fail to learn to escape from subsequent stressors, more so than rodents previously not exposed to stress. Thus, an absence of control leads to learned helplessness in novel contexts. Additional effects of such uncontrollable stress include reduced aggression, social dominance, and extinction learning as well as increased fear conditioning (*Maier and Watkins, 2005*), a cascade of negative outcomes that has been interpreted as indexing a behavioural phenotype of internalising disorders (*Maier and Seligman, 2016*). Intriguingly, much less is known about the effects conferred by heightened control. Animal work found that exposure to controllable stress (compared to no stress at all) effectively inoculates against the negative impact of a subsequent stressor (*Kubala et al., 2012*). Similar empirical approaches in humans have shown that stressor controllability modulates fear extinction learning (*Hartley et al., 2014*). To date, however, it is unclear whether heightened control can actually buffer against the effect of a subsequent stressor in humans (*Bhanji et al., 2016*; *Bhanji and Delgado, 2014*).

To address these questions, we designed a novel task (the 'Wheel Stopping' task or WS task) requiring carefully timed actions to achieve a reward or avoid reward losses that allows inducing and measuring different states of subjective control. We first show that the task has excellent psychometric properties with high internal consistency and external validity. We then show strong and consistent associations between sense of control and stress in a non-clinical adult sample (N=473, Study 1; N=295, Study 2). Manipulating control has often presented challenges given potential confounds of task difficulty, predictability (*Ligneul, 2021*), and the probability of action and reward outcomes (*Dorfman and Gershman, 2019*). Existing tasks manipulate the degree of control rather coarsely through entirely present or absent control affordances (*Dorfman and Gershman, 2019*; *Hartley et al., 2014*; *Ligneul, 2021*; *Ligneul et al., 2022*; *Raab et al., 2022*). Here, we verify that the relationship between subjective control and stress is apparent above and beyond the variance accounted for by perceived task difficulty. We then use the task to examine the effect of *heightened control* on the response to future stressors. By manipulating subjective control during the task, we show that heightened sense of control buffers against the effects of a subsequent stressor (N=295; Study 2). Given the pervasive occurrence of stress and its wide-reaching negative impact, these findings have considerable translational potential.

## Results

We first investigate the internal consistency and external validity of the task. We also use a simple computational model to relate subjective control responses to the trial-specific parameters at the individual level. We then investigate the within-task associations of subjective stress and control in Study 1 and replicate this in Study 2. Finally, in Study 2, we investigate whether experiencing heightened control from the WS task, compared to watching videos, buffers responses to a subsequent stressor.

### Internal consistency

We estimated internal consistency for the slider scale ratings by calculating the intraclass correlation coefficient (ICC) between two halves of the data (here, using the mean of odd numbered sliders and the mean of the even numbered sliders as a two-way mixed effects model, single measurement type testing for absolute agreement *Koo and Li, 2016*). In Study 1, there was high internal consistency for control ratings (ICC(A,1)=0.905 [95% CI: 0.888, 0.920], $F(472,473) = 20.2$, p<0.001) and difficulty ratings (ICC(A,1)=0.952 [95% CI: 0.943, 0.960], $F(472,473) = 40.5$, p<0.001). Internal consistency was slightly lower but still good for the stress ratings (ICC(A,1)=0.777 [95% CI: 0.633, 0.854], $F(472,28.9)=9.73$, p<0.001), likely due to the stress task resulting in the subsequent stress slider scale values falling into one of the halves. Study 2 also had good internal consistency for control ratings (ICC(A,1)=0.803 [95% CI: 0.747, 0.847], $F(200,194) = 9.26$, p<0.001), difficulty ratings (ICC(A,1)=0.717 [95% CI: 0.643, 0.778], $F(200,200) = 6.05$, p<0.001), and stress ratings (ICC(A,1)=0.878 [95% CI: 0.751, 0.930], $F(294,17.7)=20.00$, p<0.001). We also calculated the ICC using a 1st/2nd half split of the data (*Supplementary file 1*).

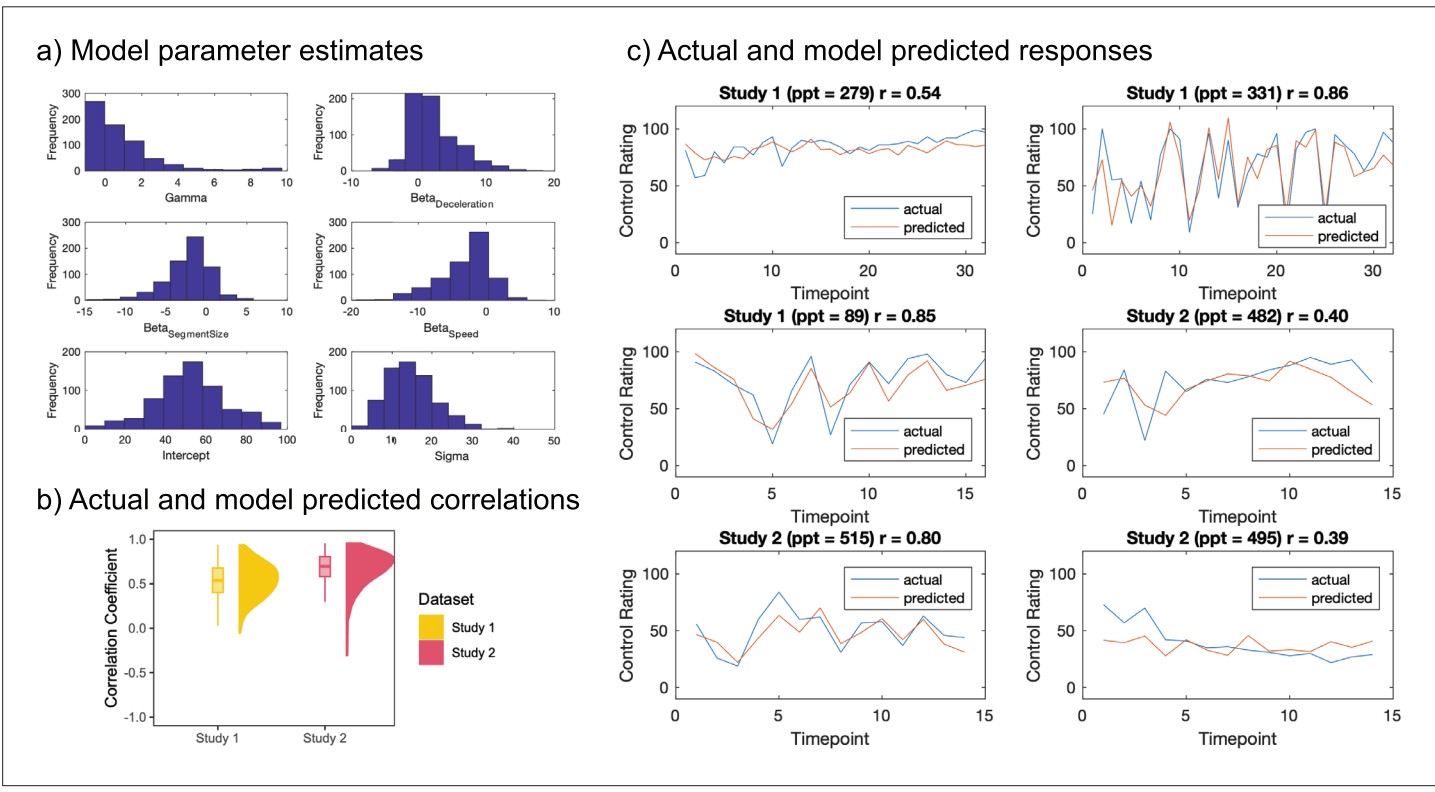

**Figure 1.** Linear model predicting sense of control from WS task parameters. (**a**) Parameter estimates across all participants who completed the WS task (n = 674). These were significantly different from 0 (one-sampled *t*-tests, FDR-corrected p values <0.001, see text for full statistical details). (**b**) Correlation coefficients between predicted and actual control ratings for both studies (Study 1 median = 0.54, Study 2 median = 0.69). Boxplots show the median and 25th and 75th percentiles. Whiskers extend to 1.5*IQR (inter-quartile range) from the quartiles. (**c**) Control rating responses over the time course of the experiment predicted from the model (red) plotted against the actual ratings (blue) for three randomly selected participants per study, with the correlation coefficient (r) per participant shown in the top right of each subplot.

The online version of this article includes the following figure supplement(s) for figure 1:

**Figure supplement 1.** Mean BIC (across all participants and both studies) for each of the 7 models with different regressors.

## Predicting sense of control from task parameters

To establish whether the Wheel Stopping task parameters governing the difficulty and hence controllability of individual trials (speed of rotation, segment size, deceleration increment) affected participants' subjective sense of control, we created a linear model relating trial-by-trial variation in task parameters to the time series of control ratings. The best model (lowest BIC, *Figure 1—figure supplement 1*) predicted control ratings from all three Wheel Stopping task parameters (deceleration increment, segment size, and speed of rotation), although all models performed similarly. The winning model contained an intercept term, a gamma term to discount previous control ratings at the current time point, and the beta weights for the three regressors. A histogram of the parameter estimates across all participants is shown in *Figure 1a*. Parameter estimates across the whole sample were significantly different from 0 (one-sampled *t*-tests, ps <0.001: Gamma $t(673)$=12.19, p<0.001, $p_{adj.}$ <0.001; Deceleration $t(673)$=17.14, p<0.001, $p_{adj.}$ <0.001; Segment Size $t(673)$=-20.71, p<0.001, $p_{adj.}$ <0.001; Speed $t(673)$=-20.80, p<0.001, $p_{adj.}$ <0.001; Intercept $t(673)$=76.82, p<0.001, $p_{adj.}$ <0.001). On average, the beta coefficients for Segment Size (mean = –2.25, SD = 2.83) and Speed (mean = –3.09, SD = 3.86) were negative, showing that a faster speed and a smaller segment size (a larger fraction parameter) resulted in lower ratings of control. On the other hand, on average, the beta weight for deceleration increment was positive (mean = 2.33, SD = 3.53), showing that the wheel stopping faster predicted higher feelings of control. The median correlation between actual and fitted control responses across the whole sample was 0.60 (Study 1=0.54, Study 2=0.69, *Figure 1b*). The actual and predicted control responses over the course of the experiment can be seen for six randomly selected participants (three per study iteration) in *Figure 1c*. Overall, this modelling approach shows that the subjective control response can be captured well from the Wheel Stopping task parameters at the individual level, and that there is individual variation among how the task parameters are weighted to predict control, as well as variation in the intercept.

## External validity

To establish the external validity of our task-based measures of subjective control and stress ratings, across both studies, we correlated participants' mean level of control and stress from the experiment with standardised questionnaire measures of anxiety, depression, social phobia and locus of control. Given that the experimental manipulation of WS task control conditions (low or high) in Study 1 was designed to influence subjective control ratings in the task, and the stressor intensity conditions (low or high) in both studies were designed to influence stress ratings in the task, these experimental conditions were included as covariates in the respective models.

We found that in both studies, higher mean subjective control during the WS task was associated with lower levels of trait anxiety (study 1: $\beta$=–0.24, p<0.001, $p_{adj.}$ <0.001; study 2: $\beta$=–0.37, p<0.001, $p_{adj.}$ <0.001), initial state anxiety (study 1: $\beta$=–0.25, p<0.001, $p_{adj.}$ <0.001; study 2: $\beta$=–0.39, p<0.001, $p_{adj.}$ <0.001), and depressive symptom severity (study 1: $\beta$=–0.50, p=0.004, $p_{adj.}$=0.007; study 2: $\beta$=–0.55, p=0.006, $p_{adj.}$=0.008). In study 2 (but not significantly in study 1), higher mean subjective control during the WS task was predicted by a more internal locus of control (study 1: $\beta$=–0.30, p=0.191, $p_{adj.}$=0.239; study 2: $\beta$=–0.80, p=0.008, $p_{adj.}$=0.008) and lower social phobia symptom severity (study 1: $\beta$=–0.06, p=0.350, $p_{adj.}$=0.350; study 2: $\beta$=–0.34, p<0.001, $p_{adj.}$ <0.001), after accounting for the WS task control condition in all Study 1 models, where the low control condition significantly predicted lower mean control ratings (ps <0.001, $ps_{adj.}$ <0.001; *Supplementary file 2A and B*, *Figure 2—figure supplement 1*). These results replicated when correlating questionnaire scores with our model-derived parameter estimates of the intercept term from predicting individuals' control ratings from the WS task parameters (*Supplementary file 2C and D*).

We found that in both studies, higher mean subjective stress in the experiment was associated with higher levels of trait anxiety (study 1: $\beta$=0.92, p<0.001, $p_{adj.}$<0.001; study 2: $\beta$=1.12, p<0.001, $p_{adj.}$ <0.001), initial state anxiety (study 1: $\beta$=1.06, p<0.001, $p_{adj.}$ <0.001; study 2: $\beta$=1.27, p<0.001, $p_{adj.}$ <0.001), depressive symptom severity (study 1: $\beta$=1.88, p<0.001, $p_{adj.}$ <0.001; study 2: $\beta$=2.21, p<0.001, $p_{adj.}$ <0.001), and social phobia symptom severity (study 1: $\beta$=0.47, p<0.001, $p_{adj.}$ <0.001; study 2: $\beta$=0.74, p<0.001, $p_{adj.}$ <0.001), and a more external locus of control (study 1: $\beta$=0.75, p=0.013, $p_{adj.}$=0.013; study 2: $\beta$=1.58, p<0.001, $p_{adj.}$ <0.001), after accounting for stressor intensity condition (*Supplementary file 2E and F*, *Figure 2—figure supplement 2*).

**Table 1.** The unique contributions of subjective control and perceived task difficulty on subjective stress during the Wheel Stopping task.

| Predictors | Subjective Stress (Study 1) | |
| --- | --- | --- |
| | Estimates (95% CI) | p |
| (Intercept) | 31.13 (25.15–37.12) | <0.001 |
| Subjective Control | –0.13 (–0.20––0.07) | <0.001 |
| Perceived Difficulty | 0.38 (0.31–0.44) | <0.001 |
| Random Effects | | |
| $\sigma^2$ | 208.61 | |
| $\tau_{00}$ | 346.20 $_{ppt}$ | |
| | 3.63 $_{timepoint}$ | |
| ICC | 0.63 | |
| N | 473 $_{ppt}$ | |
| | 4 $_{timepoint}$ | |
| Observations | 1892 | |
| Marginal $R^2$ /Conditional $R^2$ | 0.154/0.684 | |

## Stress and control

During the WS task, we measured subjective stress ratings using a slider scale, assessed every 16 blocks (80 trials) in Study 1, and every 9 blocks (45 trials) in Study 2 (**Supplementary file 5D**). We also assessed subjective control and perceived difficulty during the WS task, assessed every 2 blocks (10 trials) in both studies. To investigate how subjective control related to feelings of stress during the WS task while accounting for perceived task difficulty, we took the mean control ratings and mean difficulty ratings preceding each stress rating, enabling us to have a value of control and difficulty corresponding to each stress timepoint. For Study 1 (n=473), there were four subjective stress ratings (timepoints) during the WS task. A linear mixed effects model including random intercepts of participant and timepoint showed that lower feelings of control ($\beta$=–0.13, p<0.001) and higher perceived task difficulty ($\beta$=0.38, p<0.001) were both uniquely associated with higher subjective stress during the WS task (**Table 1**, **Figure 2**). This effect replicated in Study 2 (n=201), with three (instead of four) timepoints, such that higher subjective stress during the WS task was uniquely associated with lower subjective control ($\beta$=–0.33, p<0.001) and higher perceived task difficulty ($\beta$=0.32, p<0.001; **Supplementary file 3A**, **Figure 2**).

To further isolate the relationship between subjective control and stress separate from perceived task difficulty or objective task performance, we also included the overall win rate (percentage of trials won during the WS task) in the models. In Study 1, lower feelings of control were related to higher levels of subjective stress ($\beta$=–0.12, p<0.001) even when controlling for both win rate ($\beta$=–0.06, p=0.220) and perceived task difficulty ($\beta$=0.37, p<0.001, **Supplementary file 3B**). This also replicated in Study 2, where lower subjective control was associated with higher feelings of stress ($\beta$=–0.32, p<0.001) when controlling for perceived task difficulty ($\beta$=0.31, p<0.001) and win rate ($\beta$=–0.11, p=0.428, **Supplementary file 3A**). This suggests that there is unique variance in subjective feelings of control, separate from task performance, relevant to subjective stress.

Study 2 also investigated how the loss domain (avoid losing bonus money) and win domain (winning 1 p per correct trial) influenced feelings of stress during the task. We included subjective control and perceived difficulty in the model as known covariates that influence stress in the task and accounted for random intercepts of participant and timepoint. We found that the loss domain ($\beta$=8.99, p=0.001) uniquely predicted higher feelings of stress, along with lower feelings of subjective control ($\beta$=–0.31, p<0.001) and higher perceived task difficulty ($\beta$=0.31, p<0.001), as before (**Supplementary file 3A**).

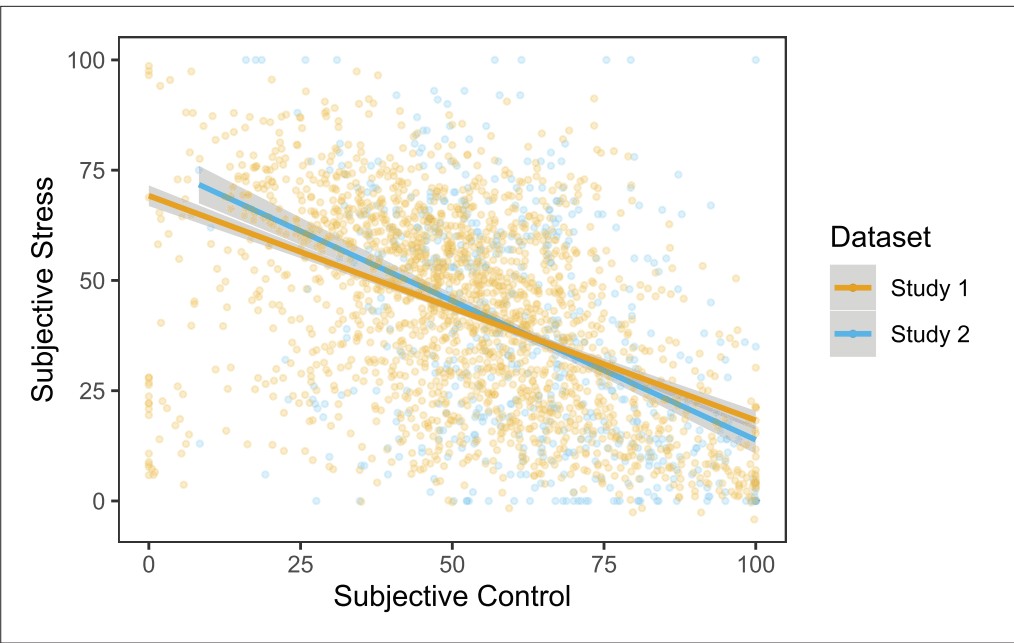

**Figure 2.** Negative association between subjective control and subjective stress during the Wheel Stopping task for both studies. Points represent raw data (Study 1: n = 473, 4 timepoints; Study 2: n = 201, 3 timepoints) and lines represent the estimated relationship from the linear mixed effects models in *Table 1* (Study 1) and *Supplementary file 3A* (Study 2), showing the relationship between subjective control and subjective stress (Study 1: $\beta$=–0.13, p<0.001; Study 2: $\beta$=–0.33, p<0.001), after accounting for perceived task difficulty and random effects of participant and timepoint. Shaded regions represent 95% confidence intervals.

The online version of this article includes the following figure supplement(s) for figure 2:

**Figure supplement 1.** Associations between mean subjective control and questionnaire measures in (**a**) Study 1, and (**b**) Study 2.

**Figure supplement 2.** Associations between mean subjective stress and questionnaire measures in (**a**) Study 1, and (**b**) Study 2.

---

The final WS stress rating in both studies occurred after a STAI-S questionnaire following the final block of the WS task. To ensure that this delay in the final stress rating did not affect the results, we repeated the analyses excluding the final timepoint. Results in both studies replicated, showing that the within-task relationship between subjective control and subjective stress was robust (*Supplementary file 3A and B*).

## Stressor controllability

As presented in the previous section, an increased sense of control related to lower levels of subjective stress during the Wheel Stopping task in both studies. We next examined in Study 2 whether this effect of heightened control generalised beyond the WS task in response to a subsequent stressor. To do so, we added a causal manipulation whereby one group of participants received a version of the WS task inducing high levels of control ratings. While high control is often contrasted to low control conditions, we were here interested in a stronger test to ask whether the experience of control itself might have a protective effect in comparison to an intervention that might similarly reduce stress by itself. We hence compared the high control task to a simple video watching condition. During the video task, the mean of individuals' valence ratings was 6.60 (SD = 1.53), where 1 represents unpleasantness and 9 represents pleasantness. A one-sided one-sampled t-test revealed this was significantly greater than the scale's midpoint (*t*(93)=10.17, p<0.001). The mean of individuals' arousal ratings was 3.94 (SD = 1.74), where 1 represents calmness and 9 represents excitement. A one-sided one-sampled t-test revealed this was significantly less than the scale's midpoint (*t*(93)=-5.91, p<0.001). These suggest that participants indeed found the videos relaxing (pleasant and calm). These tasks were followed by a stressor of high or low intensity (the anticipatory Trier Social Stress Test or recipe

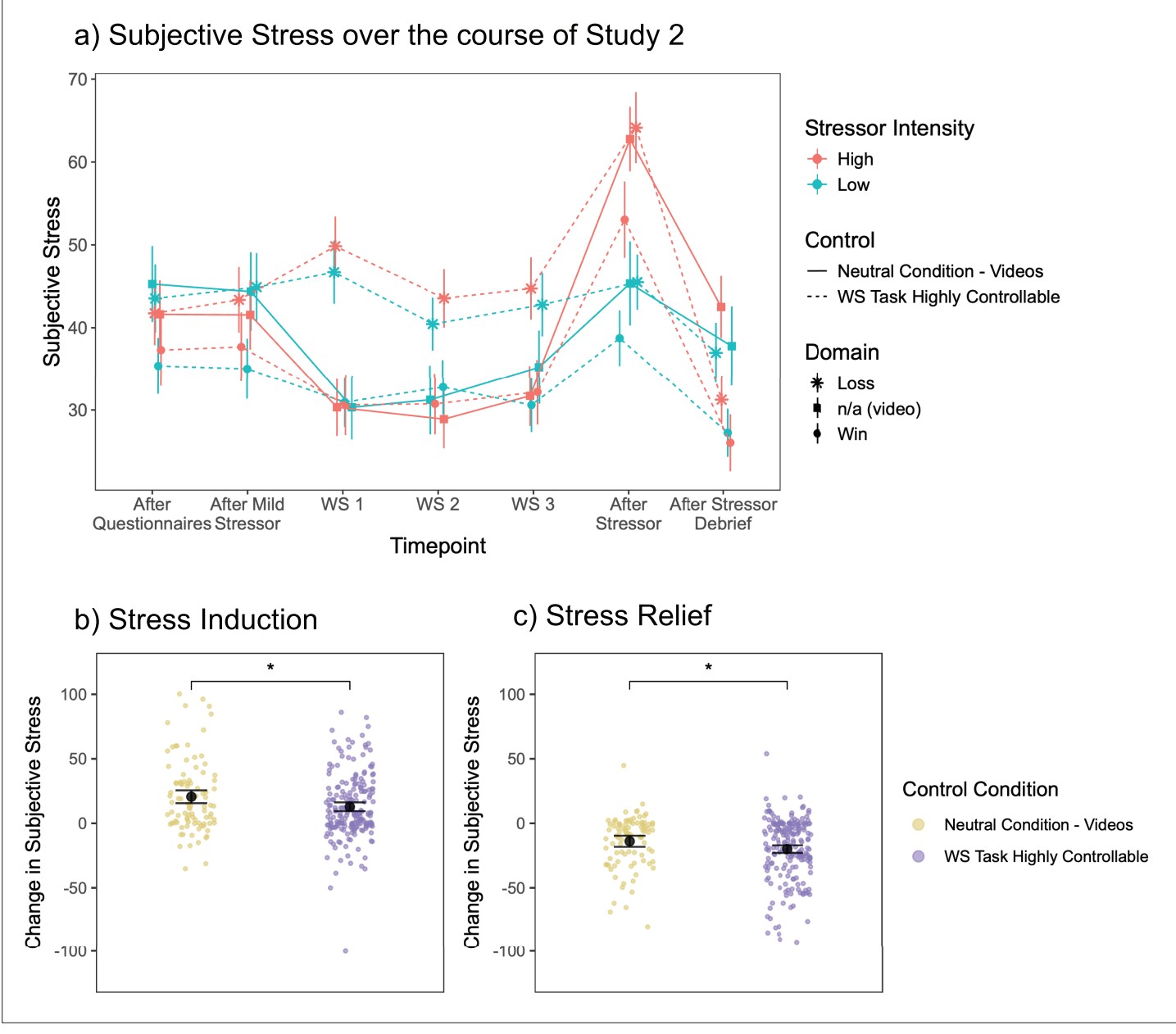

**Figure 3.** Stress induction and stress relief in Study 2. (**a**) Subjective stress ratings across the entire experiment (Study 2, n = 295) in the different experimental conditions. The point represents the mean per group, and the error bar represents standard error of the mean. Timepoints labelled WS 3, After Stressor, and After Stressor Debrief are the three timepoints isolating the stress induction and stress debrief (coded as timepoints 1, 2, 3). Jitter added to avoid overlap. (**b**) Stress Induction – the change in subjective stress from before to after the stressor. Data points show the difference between timepoints per participant. The black points show the mean estimate of the contrast between the two timepoints from the linear mixed effects models in *Table 2* (with 95% confidence intervals as error bars). The comparison is the difference between these contrasts, showing that the stress induction was lower for the high control group than for the neutral control group ($\beta$=–7.78, SE = 3.07, $t$(291)=-2.54, p=0.012). (**c**) Stress Relief – the change in subjective stress from after the stressor to after the stressor debrief. Data points show the difference between timepoints per participant. The black points show the mean estimate of the contrast between the two timepoints from the linear mixed effects models in *Table 2* (with 95% confidence intervals as error bars). The comparison is the difference between these contrasts, showing that the stress relief was greater for the high control group than for the neutral control group ($\beta$=–6.06, SE = 2.64, $t$(291)=-2.30, p=0.022).

comprehension task, respectively). The analyses in this section therefore focus on stress ratings from three timepoints: after the WS/video task, after the stressor, and after the stressor debrief, coded as timepoints 1, 2, 3, respectively. The subjective stress levels across the course of the entire experiment are shown in *Figure 3a*.

**Table 2.** Linear mixed effects models predicting subjective stress from two timepoints: before and after the stressor (stress induction, timepoints 1 and 2), and after the stressor and after the stressor debrief (stress relief, timepoints 2 and 3).

| Predictors | Stress Induction | | Stress Relief | |
|---|---|---|---|---|
| | Estimates (95% CI) | p | Estimates (95% CI) | p |
| (Intercept) | 12.99 (3.57–22.41) | 0.007 | 114.30 (102.29–126.31) | <0.001 |
| Timepoint | 20.14 (15.30–24.99) | <0.001 | −29.91 (−34.07–−25.75) | <0.001 |
| Control [Neutral] | −12.35 (−28.19–3.49) | 0.126 | −11.02 (−31.60–9.56) | 0.293 |
| Stressor Intensity [Low] | 12.89 (0.57–25.20) | 0.040 | −56.19 (−72.35–−40.02) | <0.001 |
| Domain [Loss] | 10.64 (3.73–17.54) | 0.003 | 8.19 (1.28–15.10) | 0.020 |
| Timepoint ×Control [Neutral] | 10.92 (2.38–19.46) | 0.012 | 9.65 (2.32–16.99) | 0.010 |
| Timepoint ×Stressor Intensity [Low] | −14.66 (−21.46–−7.86) | <0.001 | 19.87 (14.03–25.71) | <0.001 |
| Control [Neutral]×Stressor Intensity [Low] | 11.56 (−10.26–33.38) | 0.298 | 13.40 (−15.23–42.03) | 0.358 |
| (Timepoint ×Control [Neutral])×Stressor Intensity [Low] | −6.28 (−18.32–5.77) | 0.307 | −7.19 (−17.54–3.16) | 0.173 |
| **Random Effects** | | | | |
| $\sigma^2$ | 301.09 | | 222.45 | |
| $\tau_{00}$ | 470.40 ppt | | 510.27 ppt | |
| ICC | 0.61 | | 0.70 | |
| N | 295 ppt | | 295 ppt | |
| Observations | 590 | | 590 | |
| Marginal $R^2$ /Conditional $R^2$ | 0.131/0.661 | | 0.162/0.745 | |

## Stress induction

To examine whether subjective control induced during the task would modulate the experience of stress related to the stressor, we isolated the subjective stress ratings at timepoints 1 and 2 (after the WS/video task and after the stressor). We ran a linear mixed effects model accounting for random intercepts of participant, predicting subjective stress from control condition, stressor intensity condition, timepoint, and their interactions, and a main effect of domain condition (win or loss WS task).

There was a significant main effect of timepoint, such that subjective stress levels increased after both stressors as expected ($\beta$=20.14, SE = 2.47, p<0.001, *Table 2*). There was a significant interaction between timepoint and stressor intensity ($\beta$=−14.66, SE = 3.46, p<0.001), such that the anticipatory TSST (high stressor intensity) increased subjective stress more than the recipe comprehension task (low stressor intensity) did, as expected (estimated marginal means of subjective stress from the linear mixed effects model contrast from timepoint 1–2: high stressor intensity $\beta$=25.60, SE = 2.17, $t$(291)=11.78, $p_{adj.}$ <0.001), low stressor intensity $\beta$=7.80, SE = 2.16, $t$(291)=3.61, $p_{adj}$ <0.001, these two contrasts were significantly different from each other, $\beta$=17.8, SE = 3.07, $t$(291)=5.80, p<0.001. There was also a significant main effect of domain, such that the loss domain had overall higher subjective stress levels than the win domain ($\beta$=10.64, SE = 3.52, p=0.003), and a main effect of stressor intensity condition, as the low stressor intensity conditions had overall higher subjective stress than the high stressor intensity conditions ($\beta$=12.89, SE = 6.27, p=0.040).

There was a significant timepoint by control interaction ($\beta$=10.92, SE = 4.35, p=0.012). While there was a significant increase in stress for both control groups, post-hoc analyses revealed this increase was lower in the highly controllable WS condition than the neutral control video condition (estimated marginal means of subjective stress from the linear mixed effects model contrast from timepoint 1–2: high control $\beta$=12.8, SE = 1.73, $t$(291)=7.40, $p_{adj}$ <0.001; neutral control: $\beta$=20.6, SE = 2.53, $t$(291)=8.14, $p_{adj}$ <0.001; these two contrasts were significantly different from each other: $\beta$=−7.78, SE = 3.07, $t$(291)=-2.54, p=0.012, *Figure 3b*). There was no significant three-way interaction of timepoint by control by stressor intensity ($\beta$=−6.28, SE = 6.13, p=0.307), suggesting that the control by timepoint interaction was not dependent on the stressor intensity.

## Stress relief

To examine whether control modulates the experience of stress in response to the stressor debrief, we isolated the subjective stress ratings at timepoints 2 and 3 (after the stressor and after the stressor debrief, informing participants that they were not required to complete the stressor task as initially instructed). We ran a linear mixed effects model accounting for random intercepts of participant, predicting subjective stress from control condition, stressor intensity condition, timepoint, and their interactions, and a main effect of domain condition (win or loss WS task).

There was a significant main effect of timepoint, such that subjective stress levels decreased after the stressor debrief, as expected ($\beta$=–29.91, SE = 2.12, p<0.001, *Table 2*). There was also a significant main effect of domain, such that the loss domain had overall higher subjective stress levels than the win domain ($\beta$=8.19, SE = 3.52, p=0.020). There was also a significant main effect of stressor in both domains, such that overall, the low stressor intensity conditions had lower subjective stress ratings than the high stressor intensity conditions ($\beta$=–56.19, SE = 8.23, p<0.001).

There was a significant interaction between timepoint and stressor intensity condition ($\beta$=19.87, SE = 2.98, p<0.001, *Table 2*), such that subjective stress levels decreased after the stressor debrief more after the high intensity stressor than after the low intensity stressor (estimated marginal means of subjective stress from the linear mixed effects model contrast from timepoint 2–3: high stressor intensity $\beta$=–25.08, SE = 1.87, $t(291)$=-13.43, $p_{adj.}$ <0.001; low stressor intensity $\beta$=–8.81, SE = 1.86, $t(291)$=-4.74, $p_{adj}$ <0.001; these two contrasts were significantly different from each other: $\beta$=–16.3, SE = 2.64, $t(291)$=-6.18, p<0.001).

Regarding the main effect of interest, examining how the control condition modulates the subjective stress response over time, there was a significant two-way interaction of time and control ($\beta$=9.65, SE = 3.74 p=0.010, *Table 2*). While there was a significant decrease in subjective stress after the stressor debrief for both control groups, post-hoc analyses revealed this decrease was greater in the highly controllable WS condition than the neutral control video condition (estimated marginal means of subjective stress from the linear mixed effects model contrast from timepoint 2–3: high control $\beta$=–20.0, SE = 1.49, $t(291)$=-13.42, $p_{adj}$ <0.001; neutral control: $\beta$=–13.9, SE = 2.18, $t(291)$=-6.40, $p_{adj.}$ <0.001; these two contrasts were significantly different from each other: $\beta$=–6.06, SE = 2.64, $t(291)$=-2.30, p=0.022, *Figure 3c*). There was no significant three-way interaction of timepoint by control by stressor intensity ($\beta$=–7.19, SE = 5.27, p = 0.173), suggesting that the control by timepoint interaction was not dependent on the stressor intensity.

## Sensitivity and exploratory analyses

We conducted a series of sensitivity analyses to ensure our findings were robust. Although there were no significant differences between control groups in subjective stress immediately after the WS/video task ($t(175.6)$=1.17, p=0.244), we included participants' stress level after the WS/video task as a covariate in the stress relief analyses (*Supplementary file 4A*). The results revealed a main effect of initial stress ($\beta$=0.64, SE = 0.04, p<0.001, *Supplementary file 4A*) on the stress relief after the stressor debrief. Compared to excluding initial stress as in the original analyses (*Table 2*), there was now no longer a main effect of domain ($\beta$=0.24, SE = 2.60, p=0.093, *Supplementary file 4A*), but the inference of all other effects remained the same. Importantly, there was still a significant time by control interaction ($\beta$=9.65, SE = 3.74, p=0.010, *Supplementary file 4A*) showing that the decrease in stress after the debrief was greater in the highly controllable WS condition than the neutral control video condition, even when accounting for the initial stress level.

Our second sensitivity analysis was conducted because the experiment took longer to complete for the video condition (mean = 54.3 min, SD = 12.4 min) than the WS task condition (mean = 39.7 min, SD = 12.8 min, $t(186.19)$=−9.32, p<0.001). We therefore included the total time (in ms) as a covariate in the stress induction and stress relief analyses for Study 2. This showed that accounting for total time did not change the results of interest (*Supplementary file 4B*), further highlighting that the time by control interactions were robust.

Third, we included the interaction of domain with stressor intensity and with time to test whether the win or loss domain in the WS task significantly impacted stress induction or stress relief differently depending on stressor intensity. There were no significant effects or interactions of domain (*Supplementary file 4C*) for stress induction or stress relief, and the main effect of interest (the interaction between time and control) still held for the stress induction ($\beta$=10.20, SE = 4.99 p=0.041,

*Supplementary file 4C*), though it was no longer significant for the stress relief ($\beta$=6.72, SE = 4.28, p=0.117, *Supplementary file 4C*). This more complex model did not significantly improve model fit ($\chi^2$(3)=1.46, p=0.691) compared to our original specification (with domain as a covariate rather than an interaction) and had slightly worse fit (higher AIC and BIC) than the original model (AIC = 5477.2 vs 5472.7, BIC = 5538.5 vs 5520.8).

Finally, to test whether the loss domain was more valuable at mitigating experiences of stress than the win condition, we ran additional analyses with just the high control condition (WS task) for the stress induction and stress relief to test for an interaction of domain and time. For the stress induction, there was no significant two-way interaction of domain and time ($\beta$=−1.45, SE = 4.80, p=0.763), nor a significant three-way interaction of domain by time by stressor intensity ($\beta$=−3.96, SE = 6.74, p=0.557, *Supplementary file 4D*), suggesting that there were no differences in the stress induction dependent on domain. Similarly, for the stress relief, there was no significant two-way interaction of domain and time ($\beta$=−5.92, SE = 4.42, p=0.182), nor a significant three-way interaction of domain by time by stressor intensity interaction ($\beta$=8.86, SE = 6.21, p=0.154, *Supplementary file 4D*), suggesting that there were no differences in the stress relief dependent on the WS Task domain.

## Discussion

We designed a novel Wheel Stopping task to assess subjective sense of control and its relationship with subjective stress in a large sample of adult participants across two separate studies. Across both studies, we could show that task features designed to manipulate subjective control, namely wheel speed, segment size, and deceleration increment, were highly effective at doing so. Further, the task possessed excellent psychometric properties, including high internal consistency as well as high external validity, as demonstrated by relationships with state anxiety, locus of control, and mental health measures. We also showed that the subjective sense of control, as elicited by task parameters and experienced stress, was tightly coupled throughout the task for our participants. In a final step, we sought to causally manipulate and heighten subjective sense of control and to test its impact on subjective stress in response to a subsequent stressor. We could show that experimentally increased sense of control buffered subjectively experienced stress to a psychosocial stressor and also led to greater stress relief compared to a neutral video task.

Given the different parameter combinations of our task, we were able to elicit a considerable range in the subjective experience of control. Notably, there was a tight coupling between subjectively experienced control and subjectively experienced stress during the task. While our WS task was in and of itself not designed to elicit feelings of stress, these results are striking as clearly the variable sense of control, as elicited through changes in the achievability of obtaining instrumental rewards, was highly predictive of subjective stress levels. Importantly, during the task, we also obtained ratings of perceived task difficulty. While subjective stress was partly accounted for by task difficulty, there was also a unique relationship with perceived control. These findings demonstrate how closely subjective control tracks subjective stress independently of any stressful task properties, which in the present case were minimal and reinforce that perceived control presumably not only moderates subjective stress but impacts it directly. Mean levels of subjective sense of control during the task also correlated with a cluster of self-reports and questionnaires such as state anxiety, locus of control, trait anxiety, depression, and social phobia, replicating prior work on perceived control and mental health (*Gee et al., 2022*).

In a crucial set of experiments (Study 2), we tested whether causally manipulating and increasing participants' sense of control through our task impacts not just presently but also subsequently experienced stress. To elicit stress, we used a well-known psychosocial stressor (*Kirschbaum et al., 1993*; *Steinbeis et al., 2015*), which presently also led to a clear increase in subjective stress and subsequent relief after being debriefed. Thus, compared to participants in a neutral control condition (watching videos), participants who had been exposed to the WS task designed to give them an increased sense of control showed a reduced subjective stress response following the stressor and greater relief after the debrief. These findings demonstrate that heightened control impacts subsequent stress levels. Whereas prior work in humans has shown that experimentally induced increases in experiences of control lead to greater fear extinction, a core regulatory process (*Hartley et al., 2014*), the present findings provide direct evidence that heightened control buffers against later stress.

Stress is causally implicated in the emergence and maintenance of multiple mental health conditions across the lifespan (*Lupien et al., 2009*); however, the nature of stressors and how we respond to them significantly impacts mental health symptomatology (*Dickerson and Kemeny, 2004*; *Koss and Gunnar, 2018*). Stressors are most potent when uncontrollable, as demonstrated through decades of research on learned helplessness (*Maier and Seligman, 2016*). By implication, there has been a growing interest in how *heightened* control over stressful events may mitigate their impact and lead to resilient outcomes even in response to future stressors (*Bhanji et al., 2016*; *Bhanji and Delgado, 2014*; *Hartley et al., 2014*; *Ly et al., 2019*; *Wang and Delgado, 2021*). While some cross-species work has been able to provide some initial support for this idea (*Maier and Seligman, 2016*), clear causal evidence in humans has so far been lacking. Here, we were able to show that experimentally increasing experiences of control through our WS task and thereby heightening participants' subjective sense of control led to reductions in stress following a subsequent stressor. While we were presently unable to differentiate whether this is driven by objective or subjective dimensions of control, we speculate that, similar to other domains of mental health (*Bzdok and Dunbar, 2022*; *Danese and Widom, 2020*; *Rivenbark et al., 2020*), subjective elements of control are more impactful. This raises the possibility of targeting subjective control beliefs to reduce the likely impact of stress. Previously, similar interventions have been designed to target other types of beliefs implicated in mental health symptomatology (i.e. optimism, causal attribution; *Malouff and Schutte, 2017*; *Roesch and Weiner, 2001*). Given the high degree of comorbidity among many mental health disorders (*Caspi and Moffitt, 2018*) and the crucial role of subjective sense of control across a host of mental health disorders (*Gee et al., 2022*; *Kubala et al., 2012*; *Wade et al., 2019*), the present work implicates a highly relevant target for possible intervention.

Our set of studies is not without its limitations. As stated above, we operationalised variations in task controllability through task difficulty, while technically each of our WS iterations was controllable. Our intention was to selectively target subjective control, which we were able to achieve, while simultaneously statistically controlling for perceived task difficulty. Future work will need to find ways of operationalising controllability in a graded fashion while accounting for other confounds such as predictability (*Ligneul, 2021*). Further, similar to other task designs comparing control conditions (*Dorfman and Gershman, 2019*), these conditions were not matched for associated rewards. Previous research typically accounts for different outcomes (e.g. punishment) by yoking controllable and uncontrollable conditions (*Maier and Seligman, 2016*), although other work has manipulated the controllability of rewards by changing the reward rate (e.g. *Dorfman and Gershman, 2019*) where a decoy stimulus is rewarded 50% of the time in the low control condition but 80% in the high control condition. While our task design does not separate control from obtained reward, we are able to do so in the statistical analyses. Like with perceived difficulty, we statistically accounted for reward rate and showed that the relationship between subjective control and stress was not accounted for by reward rate, for example. Similarly, participants received feedback after every trial, and thus feedback valence may contribute to stress perception. However, given that overall win rate (which captures the feedback received during the task) did not predict stress over and above perceived difficulty or subjective control, it suggests that feedback is unlikely to relate to stress over and above difficulty. Future work will need to disentangle this further to rule out such potential confounds. Another avenue for future research would be to test how control buffers against stress when compared to a neutral control scenario of higher stress levels, akin to the loss domain in the WS Task, given that participants found the video condition generally relaxing. However, given that we found no differences dependent on domain for the stress induction in the WS Task conditions, it is possible that different versions of a neutral control condition would not impact the stress induction.

We designed a novel experimental task to track and induce subjective sense of control in a highly graded fashion. While this task possessed excellent psychometric properties, we were also able to show that momentary sense of control was tightly coupled with subjective stress levels as well as state anxiety and symptoms of mental ill health. Crucially, compared to a relaxation control condition, an experimental and task-induced increase in sense of control led to reduced subjective stress in response to a subsequent stressor, as well as greater stress relief after participants were debriefed. Our findings demonstrate a causal role of subjective control in mitigating the negative impact of stress and identify a highly valuable and potentially modifiable target for interventions aimed at reducing the detrimental consequences of stressors in everyday life.

**Table 3.** Overview of the two studies.

|  | Study 1 | Study 2 |
|---|---|---|
| N participants | 473 | 295 |
| N Conditions | 4 | 6 |
| WS Task Control Conditions | High, Low | High, Neutral (videos) |
| Stressor Intensity Conditions | High, Low | High, Low |
| Domain Conditions | Win | Win, Loss |
| Procedure Summary | Stressor then Wheel Stopping task | Mild stressor, Wheel Stopping task, then stressor |
| Questionnaire Measures Collected | STAI, PHQ, SPIN, LOC | STAI, PHQ, SPIN, LOC |

## Methods

### Design

The core task (Wheel Stopping or WS task) to manipulate subjective feelings of control in both Studies 1 and 2 required participants to stop a spinning wheel in the correct place using a keypress. All WS task variants were objectively controllable, but their difficulty varied such that some were more difficult to control. Study 1 investigated the within-task coupling of subjective stress and control, while Study 2 investigated whether experiencing a highly controllable WS task in comparison to rating affectively neutral videos (therefore termed a neutral control condition) moderated responses to a subsequent stressor of high or low intensity.

Study 1 employed a 2x2 factorial design, with two levels of control (high or low) and two levels of stressor intensity (high or low). To increase subjective stress levels during the WS task in Study 2, an additional experimental condition was added such that participants would lose the entirety of a monetary bonus if a randomly selected trial was unsuccessful (named the 'loss' domain), given the aversiveness of potential losses (*Delgado et al., 2006*; *Loewenstein et al., 2001*). An overview of the study conditions is shown in *Table 3*. As a result, for Study 2, there were six conditions made up from two levels of control (high and neutral), two levels of stressor intensity (high or low) and two levels of domain (win or loss) for the high control conditions (WS task).

### Ethics

This study was approved by UCL research ethics committee (12271/003). Electronic informed consent was obtained from all participants. Participants received an average base payment of £7.42 /hr plus the opportunity to win bonus money depending on their performance. The bonus was 1 p per correct trial in the 'win' domain conditions. In the 'loss' domain conditions, participants were assigned a £3 bonus at the start of the study, and a randomly selected trial determined whether they lost (if the trial

**Table 4.** Demographic information for the two study iterations.

|  | Study 1 | Study 2 |
|---|---|---|
| Age – mean (SD) | 30.2 (8.18) | 28.6 (4.84) |
| Female – n (%) | 241 (51.2) | 148 (50.3) |
| Nationality UK – n (%) | 377 (80.0) | 239 (81.3) |
| First language English – n (%) | 399 (84.7) | 255 (86.7) |
| Ethnicity – n (%) Asian Black Mixed White Other | - | 30 (10.20) 24 (8.19) 12 (4.08) 219 (74.49) 6 (2.05) |

Notes: Contains missing demographic data for some participants. Missing: age from 5 participants (Study 1 n=3, Study 2 n=2), sex from 5 participants (Study 1 n=3, Study 2 n=2), nationality from 10 participants (Study 1 n=7, Study 2 n=3), first language from 10 participants (Study 1 n=8, Study 2 n=2), ethnicity from all Study 1 and from Study 2 n=4.

was incorrect) or kept (if the trial was correct) the entirety of the bonus. In the video condition (Study 2, neutral control), all participants received an additional £1 bonus. Using existing Prolific screening criteria, all participants agreed they had a webcam, audio, and microphone; agreed to their video to be recorded; and agreed that they were comfortable to be deceived. This was to increase the believability of the stressor, but no webcam or audio data were recorded, and participants were debriefed that the stressor task would not happen during the experiment.

## Participants

The final sample for both studies included 768 participants located in the UK (50.8% female, mean age = 29.6 years, SD = 7.12) and was recruited online via Prolific (https://www.prolific.com/) between August 2021 and November 2022. The majority had UK Nationality (80.5%) and English as their first language (85.5%). Demographic information per study is shown in *Table 4* (see *Supplementary file 5A and B* for demographic information and questionnaire measures broken down by study and experimental condition). Note that ethnicity data was provided by Prolific only for Study 2. The sample size of around 50 participants per experimental condition was chosen as double what has been used in previous between-subjects stressor controllability work with humans (*Hartley et al., 2014*) and deemed sufficiently powered.

Twelve additional participants were recruited but excluded from the final sample, four participants due to duplicate data, two participants due to failing questionnaire attention checks and poor performance on the Wheel Stopping task (win rate below 1SD of the mean), and six due to failing less than 3/6 attention checks on the video task. See *Supplementary file 5C* for more details on these.

## Materials

### Software

The experiment was programmed using JavaScript and HTML, including plugins from jsPsych (version 6.1.0; *de Leeuw, 2015*). The experiment was hosted online using Firebase (http://firebase.google.com/).

### Wheel stopping task

To induce high or low feelings of control, we used two versions of the Wheel Stopping (WS) task: a single-press and a multi-press version (between-subjects manipulation). While all variants were technically controllable, the variations differed in difficulty making wheels subjectively easier or harder to control. In the WS task, a yellow segment spins within a blue circle, and the participants' goal is to stop the yellow segment over a red 'break zone' by pressing the 'b' key. In the single-press version, one press of the brake causes the segment to stop. How quickly the wheel stops after a keypress is determined by a stopping angle. For example, a stopping angle of $\pi/2$ means the segment stops a quarter of a rotation after the brake is pressed (since a whole rotation of the circle is $2\pi$ radians). The deceleration increment for a given trial in the single-press condition was calculated as $\frac{\sqrt{speed}}{2 \times stoppingAngle}$. For the multi-press version, pressing the brake increases the brake *strength* in an incremental way. The greater strength of the brake (here, how many times the 'b' key is pressed) the sooner it stops. The stopping parameter in the multi-press is the deceleration increment. The speed and segment width of the wheel were also manipulated. Variations of difficulty were implemented by modifying a combination of segment speed, width, and deceleration (within-subjects manipulation). As well as a different braking procedure, the low control version also had higher speeds, but the segment sizes were the same for both high and low control versions. The order in which the parameter combinations (speed, segment width, deceleration increment) were presented was randomised across participants (using JavaScript's 'Math.random()' function), except in Study 2, in which the parameters became increasingly easier to elicit increasing feelings of control (see *Supplementary file 5D* for task parameters). Note that Study 1 included high and low control versions of the WS task, whereas Study 2 only included the high control version of the WS task. After each trial, participants were shown written feedback on screen as to whether the segment had successfully stopped on the red zone (or not), and the associated reward (or lack of). See *Supplementary file 5D* for details.

## Stressor

### High intensity

To induce increased levels of stress (between-subjects manipulation), we ran a modified version of the Trier Social Stress Test (TSST; *Kirschbaum et al., 1993*; *Steinbeis et al., 2015*). Participants were instructed that they had 10 min to mentally prepare a five-minute speech describing why they would be a good candidate for their ideal job. Participants were told the speech would be video recorded and reviewed by a panel of judges trained in public speaking to evaluate its clarity, style, and how persuasive the presentation was, and performance would be ranked relative to other participants. Participants did not actually perform the speech, so we refer to the high intensity stressor as an anticipatory TSST. Prior research has found TSST anticipation to elicit both psychological and physiological stress responses (*Nasso et al., 2019*; *Schlatter et al., 2021*; *Steinbeis et al., 2015*), suggesting that the task anticipation would be a valid stress induction despite participants not performing the speech task. Moreover, prior research has validated the use of remote TSST in online settings (*DuPont et al., 2022*; *Meier et al., 2022*), including evidence that the speech preparation phase (online) was related to increased heart rate and blood pressure compared to controls (*DuPont et al., 2022*).

### Low intensity

To induce comparably low(er) levels of stress, participants read a recipe of how to bake a loaf of bread and were asked to remember as much information as possible for 10 min before completing a short quiz about the recipe. The experiment automatically continued after 10 min for both conditions.

## Stressor debrief

For both stressor intensity conditions, participants were told that they did not have to complete the stressor task. Immediately before the stressor debrief, participants rated how stressed they were feeling, and immediately after the stressor debrief, participants completed a STAI-S questionnaire and rated how stressed they were feeling.

## Preliminary mild stressor

For Study 2, it was announced to participants prior to the WS task that they would be asked to prepare a short task which would be video recorded and evaluated by the research team. This was done to induce mildly elevated levels of anticipatory stress during the WS task.

## Video task

For the 'neutral control' condition (i.e. neither high nor low control, as used in Study 1) used in Study 2, a video task was used. This contained 12 videos, each 1 min 15 s in length showing a montage of landscape scenes. After every video, participants completed the Self-Assessment Manikin (SAM; *Bradley and Lang, 1994*) to measure valence, arousal, and dominance. Participants also answered attention check questions about the videos (e.g. 'In the video, what is the weather like in the first scene? (a) rainy, (b) sunny, (c) cloudy'). There were three blocks, each containing four videos and two attention checks. This was chosen as a control condition to avoid control affordances typical of active tasks. The chosen videos were affectively neutral but engaging. Given the task itself is designed to be relaxing, it is a strong control for investigating effects on stress.

## Slider rating measures

During the WS task, we asked how participants were feeling using three different slider rating scales. One to measure control: 'How in control do you feel right now?' from 'very out of control' to 'very in control'; one to assess their levels of stress: 'How much stress are you currently experiencing?' from 'very little stress' to 'a lot of stress'; and one to measure perceived task difficulty: 'How difficult are you finding the task right now?' from 'not difficult at all' to 'very difficult'. All responses corresponded to values 0–100, although no numerals or points along the line were visible to participants. For both studies, slider rating scales were presented in the same order for all participants (*Supplementary file 5D*).

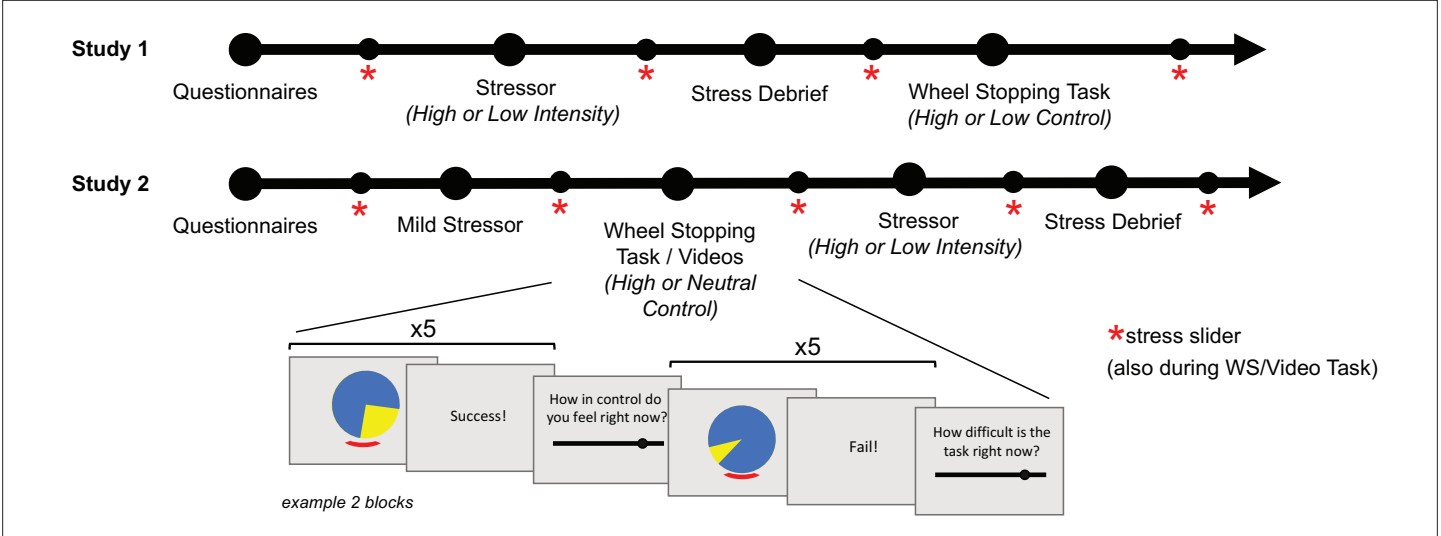

**Figure 4.** Summary of task procedure for Studies 1 and 2. The arrow shows the overall procedure with a simplified schematic of two Wheel Stopping task blocks. Participants also rated subjective stress levels on similar slider rating scales (not shown here) during the WS/Video tasks.

## Questionnaires

### Locus of control

To assess general beliefs about control, participants completed the 29-item Locus of Control (LOC) questionnaire (*Rotter, 1966*). Higher scores indicate more external locus of control. Due to an error, Study 1 was missing 1 item of the questionnaire (item 20).

### Mental health

The State-Trait Anxiety Inventory (STAI; *Spielberger et al., 1999*) was used to assess anxiety, which includes 20 items for the state subscale (STAI-S) and 20 for the trait subscale (STAI-T). We also included the 17-item Social Phobia Inventory (SPIN; *Connor et al., 2000*) to measure social phobia, and the 9-item Patient Health Questionnaire (PHQ; *Kroenke et al., 2001*) to measure depressive symptom severity.

### Attention checks

Attention checks were included for questionnaires longer than 10 items (e.g. questions such as 'select this option', 'I am paying attention and answering truthfully' included within the questionnaires). Participants who failed an attention check were given a missing value for that questionnaire.

### Procedure

We ran each study and each condition sequentially on Prolific. Researchers knew participants' assignment to an experimental condition, but participants were not aware (single-blinded). After reading the information sheet and signing the electronic consent form, participants completed the questionnaires, stressor task, WS task (or video task for Study 2 neutral control condition), and answered additional stress ratings and STAI-S throughout the experiment. The stressor task was run *before* the WS task in Study 1 and *after* the WS task for Study 2 (*Figure 4*). Given that we were interested in the experience of control on stress *buffering*, the stressor task is therefore only analysed for Study 2. A subset of Study 1 (n=175, 37%) did not have all questionnaires collected (included STAI, but not LOC, PHQ, SPIN), and due to a technical error, control ratings were provided every four blocks and difficulty sliders after all other blocks, while control and difficulty ratings alternated to be provided every two blocks in the remaining sample of Study 1 and for Study 2. The precise order of tasks and detailed information for the two studies is shown in *Supplementary file 5D*.

## Data analysis

Statistical analyses were performed in R version 4.2.2 (*R Development Core Team, 2022*), including packages from 'tidyverse' (*Wickham et al., 2019*) for data wrangling and plots, 'rstatix' (*Kassambara, 2023b*) for simple statistical models, 'sjPlot' (*Lüdecke, 2024*) for making tables of model estimates, and 'ggpubr' (*Kassambara, 2023a*) for arranging plots. Linear mixed effects models were fit by restricted maximum likelihood (REML) with *t*-tests using Satterthwaite's method using the R package 'lme4' (*Bates et al., 2015*). Estimates reported from linear models are unstandardised. For post-hoc analyses, estimated marginal means were investigated using the 'emmeans' package (*Lenth, 2022*), using Kenward-Roger degrees-of-freedom method. We corrected for multiple comparisons using the Benjamini-Hochberg (False Discovery Rate) procedure. When manually adjusting p values (outside of inbuilt function options), we input *p* values to be corrected into the 'p.adjust' R function. In the instance that initial unadjusted p values were non-exact (e.g.<0.0001), we used the upper limit as the estimate (e.g. 0.0001) for adjustment. Internal consistency for the slider scale ratings used the R package 'irr' (*Gamer et al., 2019*). Questionnaire scoring was completed using custom functions in MATLAB (*The MathWorks Inc, 2023b*). The linear model predicting control ratings based on Wheel Stopping task parameters was coded and run in MATLAB and used the function 'fminunc' from the Optimization Toolbox (*The MathWorks Inc, 2023a*) for optimisation. We tested seven different models with different numbers and combinations of regressors and calculated the Bayesian Information Criterion (BIC) to choose the model with the lowest BIC (see *Figure 1—figure supplement 1*). To note, the segment size value from the trial-by-trial WS task data determines the size of the blue space around the yellow segment, and thus a greater value represents a smaller segment. Deceleration increments were log-transformed to better fit a linear model. Regressors were standardised, and we found parameters that maximised the posterior probability. The final model predicted control responses at each timepoint (every WS trial) from an intercept, a gamma term that discounted the previous control rating's influence on the current timepoint, and three beta coefficients for the trial's speed, segment size, and deceleration increment (see *Equations 1–3*).

$$\hat{x}_i^t = x_i^t + \gamma\hat{x}_i^{t-1}, 2 < t < N \tag{1}$$

$$\hat{y}_{pred}^t = \sum_i \beta_i \hat{x}_i^t + \beta_0, i \in \{deceleration, segment\ size,\ speed\} \tag{2}$$

$$l = -\sum_{t=2}^{N} \left(y_{ratings}^t - \hat{y}_{pred}^t\right)^2 - \sum_i \frac{(\beta_i - \mu)^2}{\nu}, \mu = 0, \nu = 1 \tag{3}$$

where $i$ indicates parameter (deceleration, segment size, and speed), $t$ indicates trial and $x_i^t$ is the Z-scored value of task parameter $i$ on trial $t$. $x$ is an exponentially smoothed average of the task parameter values. $\beta_i$ is the weight given to the task parameter value in the regression. $\mu$ and $\nu$ are prior parameters to mildly regularise estimation of betas.

## Additional information

### Competing interests

Quentin JM Huys: has obtained fees and options for consultancies for Aya Technologies and Alto Neuroscience. The other authors declare that no competing interests exist.

### Funding

| Funder | Grant reference number | Author |
|---|---|---|
| Wellcome Trust | 10.35802/218497 | Jennifer C Fielder |
| Economic and Social Research Council | ES/V013505/1 | Nikolaus Steinbeis |
| Jacobs Foundation | | Nikolaus Steinbeis |

| Funder | Grant reference number | Author |
|---|---|---|
| University College London Hospitals Biomedical Research Centre | | Quentin JM Huys |
| Alexander von Humboldt-Stiftung | | Nikolaus Steinbeis |

The funders had no role in study design, data collection and interpretation, or the decision to submit the work for publication. For the purpose of Open Access, the authors have applied a CC BY public copyright license to any Author Accepted Manuscript version arising from this submission.

## Author contributions

Jennifer C Fielder, Conceptualization, Formal analysis, Investigation, Visualization, Writing – original draft, Writing – review and editing; Jinyu Shi, Formal analysis, Investigation, Methodology, Writing – review and editing; Daniel McGlade, Writing – review and editing; Quentin JM Huys, Conceptualization, Formal analysis, Supervision, Methodology, Writing – original draft, Writing – review and editing; Nikolaus Steinbeis, Conceptualization, Supervision, Funding acquisition, Methodology, Writing – original draft, Writing – review and editing

## Author ORCIDs

Jennifer C Fielder ![ORCID] https://orcid.org/0000-0002-5360-0494
Jinyu Shi ![ORCID] https://orcid.org/0000-0002-0111-2050
Daniel McGlade ![ORCID] https://orcid.org/0009-0005-8864-9692
Quentin JM Huys ![ORCID] https://orcid.org/0000-0002-8999-574X
Nikolaus Steinbeis ![ORCID] https://orcid.org/0000-0001-8650-4725

## Ethics

Human subjects: This study was approved by UCL research ethics committee (12271/003). Electronic informed consent and consent to publish was obtained from all participants.

Reviewer #1 (Public review): https://doi.org/10.7554/eLife.105025.3.sa1
Reviewer #2 (Public review): https://doi.org/10.7554/eLife.105025.3.sa2
Reviewer #3 (Public review): https://doi.org/10.7554/eLife.105025.3.sa3
Author response https://doi.org/10.7554/eLife.105025.3.sa4

# Additional files

## Supplementary files

Supplementary file 1. ICC results using a 1st/2nd half split of the data.

Supplementary file 2. Associations between questionnaires and task based measures. (A) Associations between questionnaire scores and mean task-level subjective control in Study 1, with WS control condition included as a covariate in the linear model. Adjusted p values ($p_{adj}$.) are FDR corrected p values given we ran five different models. (B) Associations between questionnaire scores and mean task-level subjective control in Study 2. The control condition was not included as a covariate in the linear model because the WS task was only presented in High Control. Adjusted p values ($p_{adj}$.) are FDR corrected p values given we ran five different models. (C) Associations between questionnaire scores and estimated intercept parameter from the computational model predicting control from WS task parameters in Study 1, with WS task control condition included as a covariate in the linear model. Adjusted p values ($p_{adj}$.) are FDR corrected p values given we ran five different models. (D) Associations between questionnaire scores and estimated intercept parameter from the computational model predicting control from WS task parameters in Study 2. Adjusted $p$ values ($p_{adj}$.) are FDR corrected $p$ values given we ran five different models. (E) Associations between questionnaire scores and mean task-level stress ratings, with external stressor intensity condition included as a covariate in the linear model for Study 1. Adjusted p values ($p_{adj}$.) are FDR corrected p values given we ran five different models. (F) Associations between questionnaire scores and mean task-level stress ratings, with external stressor intensity condition included as a covariate in the linear model for Study 2. Adjusted p values ($p_{adj}$.) are FDR corrected p values given we ran five models.

Supplementary file 3. Additional analyses testing the association between subjective control and subjective stress for both studies. (A) Relationship between subjective control, perceived difficulty and subjective stress during the WS Task in Study 2, also when removing the final WS timepoint and including Domain, or when including win rate. Predicted values from the leftmost column (Subjective Stress) model are presented in *Figure 2*. (B) Excluding the final timepoint to investigate the effects of control, difficulty and stress during the WS Task for Study 1 (left-hand model). Including all timepoints (as original model) and additionally including overall win rate as a covariate for Study 1 (right hand model).

Supplementary file 4. Sensitivity and exploratory analyses for stress induction and stress relief. (A) Linear mixed effects model for the stress relief when including initial stress level (after the WS/video task) as a covariate, predicting subjective stress from two timepoints: after the stressor and after the stressor debrief (timepoints 2 and 3). (B) Linear mixed effects models including total experiment time as a covariate, predicting subjective stress from two timepoints: before and after the stressor (stress induction, timepoints 1 and 2), and after the stressor and after the stressor debrief (stress relief, timepoints 2 and 3). (C) Linear mixed effects models including the interactions with Domain (rather than just as a covariate in the main analyses), predicting subjective stress from two timepoints: before and after the stressor (stress induction, timepoints 1 and 2), and after the stressor and after the stressor debrief (stress relief, timepoints 2 and 3). (D) Linear mixed effects models predicting subjective stress from two timepoints: before and after the stressor (stress induction, timepoints 1 and 2), and after the stressor and after the stressor debrief (stress relief, timepoints 2 and 3) in just the high control (WS task) condition, to test for the interactions with domain.

Supplementary file 5. Additional methodological details. (A) Descriptive statistics across the 4 conditions from Study 1. (B) Descriptive statistics across the 6 conditions in Study 2. Given that the Study 2 analyses compared group differences, we assessed group differences in demographic and questionnaire measures using a one-way ANOVA for continuous variables or a Chi-squared test for categorical variables. (C) Additional information about excluded participants. (D) Methodological details for both studies.

MDAR checklist

## Data availability

All data and analysis code are publicly available on the Open Science Framework at https://doi.org/10.17605/OSF.IO/39JSC.

The following dataset was generated:

| Author(s) | Year | Dataset title | Dataset URL | Database and Identifier |
|---|---|---|---|---|
| Fielder J | 2025 | Sense of control buffers against stress | https://doi.org/10.17605/OSF.IO/39JSC | Open Science Framework, 10.17605/OSF.IO/39JSC |

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
