## [Editor Report · eLife Assessment]

This **important** research addresses the effects of subjective control and task difficulty on experienced stress using a novel behavioral task administered on the same day in two large online samples. **Convincing** evidence is provided, establishing the internal and external task validity of the task, as well as a relationship between the sense of control and task difficulty, with individual differences in relevant mental health constructs. Evidence for the specificity of the link between control and stress would be more substantial if the design had not conflated control and reward rate. This work will be of interest to psychologists and clinicians studying the concepts of controllability, stress, and psychopathology.

---

## [Referee Report · Reviewer #1 (Public review)]

Summary:

This work investigated how the sense of control influences perceptions of stress. In a novel "Wheel Stopping" task, the authors used task variations in difficulty and controllability to measure and manipulate perceived control in two large cohorts of online participants. The authors first demonstrate that their behavioral task exhibits good internal consistency and external validity, indicating that perceived control during the task is linked to relevant measures of anxiety, depression, and locus of control. Most importantly, manipulating controllability in the task resulted in reduced subjective stress, demonstrating a direct impact of control on stress perception. However, this work has some minor limitations to this work due to the design of the stressor manipulations/measurements and the necessary logistics associated with online versus in-person stress studies.

Nevertheless, this research adds to our understanding of when and how control can influence the effects of stress and has particular relevance for mental health interventions.

Strengths:

The primary strength of this research is the development of a unique and clever task design that can reliably and validly elicit variations in beliefs about control. Impressively, higher subjective control in the task was associated with decreased psychopathology measures such as anxiety and depression in a non-clinical sample of participants. In addition, the authors found that lower control and higher task difficulty led to higher perceived stress, suggesting that the task can reliably manipulate perceptions of stress. Prior tasks have not included both controllability and difficulty in this manner and have not directly tested the direct influence of these factors on incidental stress, making this work both novel and important for the field.

Weaknesses:

One minor weakness of this research is the validity of the online stress measurements and manipulations. In this study, the authors measure subjective stress via self-report both during the task and after either a Trier Social Stress Test (high-stress condition) or a memory test (low-stress condition). One concern is that these stress manipulations were really "threats" of stress, where participants never had to complete the stress tasks (i.e., recording a speech for judgment). While this is not unusual for an in-lab study and can reliably elicit substantial stress/anxiety, in an online study, there is a possibility for communication between participants (via online forums dedicated to such communication), which could weaken the stress effects. That said, the authors did find sensible increases and decreases in perceived stress between relevant time points; however, future work could improve upon this design by including more comprehensive stress manipulations and by measuring implicit physiological signs of stress.

Comments on revisions:

I appreciate the authors' responses to my comments and concerns. I have decided not to make changes to my public review, as I believe it remains relevant and fair after revisions.

---

## [Referee Report · Reviewer #2 (Public review)]

Summary:

The authors have developed a behavioral paradigm to experimentally manipulate the sense of control experienced by participants by varying the level of difficulty in a wheel-stopping task. In the first study, this manipulation is tested by administering the task in a factorial design with two levels of controllability and two levels of stressor intensity to a large number of participants online, while simultaneously recording subjective ratings of perceived control, anxiety, and stress. In a second study, the authors employed the wheel stopping task to induce a high sense of controllability and investigate whether this manipulation buffers the response to a subsequent stress induction when compared to a neutral task, such as watching pleasant videos.

Strengths:

(1) The authors validate a method to manipulate stress.

(2) The authors use an experimental manipulation to induce an enhanced sense of controllability to test its impact on the response to stress induction.\

(3) The studies involved big sample sizes.

Weaknesses:

(1) The study was not preregistered.

(2) The control manipulation is conflated with task difficulty and, therefore, the reward rate. In the revised version of the manuscript, the authors perform statistical analysis to demonstrate that the relationship between perceived level of control and subjective stress remains robust after the inclusion of win rate in the model. This analysis strengthens the authors's claims, but the evidence would more substantial if the design did not conflate reward rate and control. The authors properly discuss this issue in the revised manuscript.

This study will be of interest to psychologists and cognitive scientists who are interested in understanding how controllability and its subjective perception influence how people respond to stress exposure. The demonstration that an increased sense of control buffers/protects against subsequent stress is important and may trigger further studies to characterize this phenomenon better. However, beyond the highlighted weaknesses, the current study only studied the effect of stress induction consequent to the performance of the WS task on the same day, and its generalizability is not warranted.

---

## [Referee Report · Reviewer #3 (Public review)]

Summary:

This is an interesting investigation on the benefits of perceiving control and its impact on the subjective experience of stress. To assess the subjective sense of control, the authors introduce a novel wheel stopping (WS) task where control is manipulated via size and speed to induce conditions of low and high control. The authors demonstrate that the subjective sense of control is associated with experienced subjective stress and individual differences related to mental health measures. In a second experiment, they further demonstrate that an increased sense of control buffers subjective stress induced by a trier social stress manipulation, more so than a typical stress-buffering mechanism of watching neutral/calming videos.

Strengths:

Several strengths of the manuscript can be highlighted. For instance, the paper introduces a new paradigm and a clever manipulation to test a significant and important question. Additionally, it is a well-powered investigation that allows for confidence in replicability and demonstrate both high internal consistency and high external validity, along with an interesting set of individual difference analyses. Finally, the results are quite interesting and support prior literature, while also making a significant contribution to the field in understanding the benefits of perceiving control.

Weaknesses:

The authors have addressed all my queries, and I believe the revised paper has been improved and will make an important contribution to the literature.

---

## [Author Response]

The following is the authors’ response to the previous reviews.

**Reviewer #1 (Public review):**
Summary:This work investigated how the sense of control influences perceptions of stress. In a novel "Wheel Stopping" task, the authors used task variations in difficulty and controllability to measure and manipulate perceived control in two large cohorts of online participants. The authors first show that their behavioral task has good internal consistency and external validity, showing that perceived control during the task was linked to relevant measures of anxiety, depression, and locus of control. Most importantly, manipulating controllability in the task led to reduced subjective stress, showing a direct impact of control on stress perception. However, this work has minor limitations due to the design of the stressor manipulations/measurements and the necessary logistics associated with online versus in-person stress studies.Nevertheless, this research adds to our understanding of when and how control can influence the effects of stress and is particularly relevant to mental health interventions.

We thank the reviewer for their clear and accurate summary of the findings.

Strengths:The primary strength of this research is the development of a unique and clever task design that can reliably and validly elicit variations in beliefs about control. Impressively, higher subjective control in the task was associated with decreased psychopathology measures such an anxiety and depression in a non-clinical sample of participants. In addition, the authors found that lower control and higher difficulty in the task led to higher perceived stress, suggesting that the task can reliably manipulate perceptions of stress. Prior tasks have not included both controllability and difficulty in this manner and have not directly tested the direct influence of these factors on incidental stress, making this work both novel and important for the field.

We thank the reviewer for their positive comments.

Weaknesses:One minor weakness of this research is the validity of the online stress measurements and manipulations. In this study, the authors measure subjective stress via self-report both during the task and also after either a Trier Social Stress Test (high-stress condition) or a memory test (low-stress condition). One concern is that these stress manipulations were really "threats" of stress, where participants never had to complete the stress tasks (i.e., recording a speech for judgment). While this is not unusual for an in-lab study and can reliably elicit substantial stress/anxiety, in an online study, there is a possibility for communication between participants (via online forums dedicated to such communication), which could weaken the stress effects. That said, the authors did find sensible increases and decreases of perceived stress between relevant time points, but future work could improve upon this design by including more complete stress manipulations and measuring implicit physiological signs of stress.

We thank the reviewer for urging us to expand on this point. The reviewer is right that stress was merely anticipatory and is in that sense different to the canonical TSST. However, there are ample demonstrations that such anticipatory stress inductions are effective at reliably eliciting physiological and psychological stress responses (e.g. Nasso et al., 2019; Schlatter et al., 2021; Steinbeis et al., 2015). Further, there is evidence that online versions of the TSST are also effective (DuPont et al., 2022; Meier et al., 2022), including evidence that the speech preparation phase conducted online was related to increases in heart rate and blood pressure (DuPont et al., 2022). Importantly, and as the reviewer notes in relation to our study specifically, the anticipatory TSST had a significant impact on subjective stress in the expected direction demonstrating that it was effective at eliciting subjective stress. We have elaborated further on this in our manuscript (pages 8 and 9) as follows:

“Prior research has found TSST anticipation to elicit both psychological and physiological stress responses [37-39], suggesting that the task anticipation would be a valid stress induction despite participants not performing the speech task. Moreover, prior research has validated the use of remote TSST in online settings [40, 41], including evidence that the speech preparation phase (online) was related to increased heart rate and blood pressure compared to controls [40].”

**Reviewer #2 (Public review):**
Summary:The authors have developed a behavioral paradigm to experimentally manipulate the sense of control experienced by the participants by changing the level of difficulty of a wheel-stopping task. In the first study, this manipulation is tested by administering the task in a factorial design with two levels of controllability and two levels of stressor intensity to a large number of participants online while simultaneously recording subjective ratings on perceived control, anxiety, and stress. In the second study, the authors used the wheel-stopping task to induce a high sense of controllability and test whether this manipulation buffers the response to a subsequent stress induction when compared to a neutral task, like looking at pleasant videos.

We thank the reviewer for their accurate summary.

Strengths:(1) The authors validate a method to manipulate stress.(2) The authors use an experimental manipulation to induce an enhanced sense of controllability to test its impact on the response to stress induction.(3) The studies involved big sample sizes.

We thank the reviewer for noting these positive aspects of our study.

Weaknesses:(1) The study was not preregistered.

This is correct.

(2) The control manipulation is conflated with task difficulty, and, therefore the reward rate. Although the authors acknowledge this limitation at the end of the discussion, it is a very important limitation, and its implications are not properly discussed. The discussion states that this is a common limitation with previous studies of control but omits that many studies have controlled for it using yoking.

We agree that these are very important issues to consider in the interpretation of our findings. It is important to note, that while our task design does not separate these constructs, we are able to do so in our statistical analyses. For example, our measure of perceived difficulty was included in analyses assessing the fluctuations in stress and control in which subjective control still had a unique effect on the experience of stress over and above perceived difficulty, suggesting that subjective control explains variance in stress beyond what is accounted for by perceived difficulty. Similarly, we have also included additional analyses in which we include the win rate (i.e. percentage of trials won) as a covariate when assessing the relationship between subjective control, perceived difficulty and subjective stress, in which subjective control and perceived difficulty still uniquely predict subjective stress when controlling for win rate. This suggests that there is unique variance in subjective control, separate from perceived task difficulty and win rate that is relevant to stress. We have included these analyses (page 16 of manuscript) as follows:

“To further isolate the relationship between subjective control and stress separate from perceived task difficulty or objective task performance, we also included the overall win rate (percentage of trials won during the WS task) in the models. In Study 1, lower feelings of control were related to higher levels of subjective stress (β = -0.12, p<.001) even when controlling for both win rate (β = -0.06, p = .220) and perceived task difficulty (β = 0.37, p<.001, Table S10). This also replicated in Study 2, where lower subjective control was associated with higher feelings of stress (β = -0.32, p<.001) when controlling for perceived task difficulty (β = 0.31, p<.001) and win rate (β = -0.11, p=.428, Table S11). This suggests that there is unique variance in subjective feelings of control, separate from task performance, relevant to subjective stress.”

As well as expanding on this in the Discussion (pages 27 and 28) as follows:

“While our task design does not separate control from obtained reward, we are able to do so in the statistical analyses. Like with perceived difficulty, we statistically accounted for reward rate and showed that the relationship between subjective control and stress was not accounted for by reward rate, for example. Similarly, participants received feedback after every trial, and thus feedback valence may contribute to stress perception. However, given that overall win rate (which captures the feedback received during the task) did not predict stress over and above perceived difficulty or subjective control, it suggests that feedback is unlikely to relate to stress over and above difficulty. Future work will need to disentangle this further to rule out such potential confounds.”

Further, in terms of the wider literature on these issues, we have added more to this point in our discussion, especially in relation to previous literature that also varies control by reward rate (e.g. Dorfman & Gershman, 2019, who use a reward rate of 80% in high control conditions and 50% in low control conditions). This can be found in the manuscript on page 27 as follows:

“Previous research typically accounts for different outcomes (e.g. punishment) by yoking controllable and uncontrollable conditions [3] though other work has manipulated the controllability of rewards by changing the reward rate [for example 30] where a decoy stimulus is rewarded 50% of the time in the low control condition but 80% in the high control condition.”

(3) The methods are not always clear enough, and it is difficult to know whether all the manipulations are done within-subjects or some key manipulations are done between subjects.

We have added more information in the methods section (page 8) clarifying withinsubject manipulations (WS task parameters) and between-subject manipulations (stressor intensity task, WS task version in Study 1, and WS task/video task in Study 2). Additionally, as recommended by Reviewer 1, we have provided more information in the methods section and Table S3 regarding the details of on-screen written feedback provided to participants after each trial of the WS Task.

(4) The analysis of internal consistency is based on splitting the data into odd/even sliders. This choice of data parcellation may cause missed drifts in task performance due to learning, practice effects, or tiredness, thus potentially inflating internal consistency.

We agree that this can indeed be an issue, though drift is likely to be present in any task including even in mood in resting-state (Jangraw et al., 2023). To respond to this specific point, we parcellated the timepoints into a 1^st^/2^nd^ half split and report the ICC in the supplementary information. While values are lower, indeed likely due to systematic drifts in task performance as participants learn to perform the task (especially for Study 2 since the order of parameters were designed to get easier throughout the experiment), the ICC values are still high. Control sliders: Study 1 = 0.82, Study 2: = 0.68; Difficulty sliders: Study 1: = 0.84, Study 2 = 0.57; Stress sliders: Study 1 = 0.45, Study 2 = 0.71. As seen, the lowest ICC is for stress sliders in Study 1. This may be because the first 3 sliders (included in the 1^st^ half split) were all related to the stress task (initial, post-stress, task, post-debrief) and the final 4 sliders (in the 2^nd^ half split) were the three sliders during the WS task and shortly afterwards.

(5) Study 2 manipulates the effect of domain (win versus loss WS task), but the interaction of this factor with stressor intensity is not included in the analysis.

We agree that this would be a valuable analysis to include. We have run additional analyses (section Sensitivity and Exploratory Analyses, pages 24 and 25), testing the interaction of Domain (win or loss) with stressor intensity (and time) when predicting the stress buffering and stress relief effects. This revealed no significant main effects of domain or interactions including domain, suggesting that domain did not impact the stress induction or relief differently depending on whether it was followed by the high or low stressor intensity condition. While the control by time interaction (our main effect of interest) still held for stress induction in this more complex model, the control by time interaction did not hold for the stress relief. However, this more complex model did not provide a better fit for the data, motivating us to continue to draw conclusions from the original model specification with domain as a covariate (rather than an interaction).

We outline these analyses on page 24 of the manuscript, as follows:

“Third, we included the interaction of domain with stressor intensity and with time, to test whether the win or loss domain in the WS task significantly impacted stress induction or stress relief differently depending on stressor intensity. There were no significant effects or interactions of domain (Table S14) for stress induction or stress relief, and the main effect of interest (the interaction between time and control) still held for the stress induction (β = 10.20, SE=4.99 p=.041, Table S14), though was no longer significant for the stress relief (β = 6.72, SE=4.28, p=.117, Table S14). This more complex model did not significantly improve model fit (χ^²^(3) = 1.46, p=.691) compared to our original specification (with domain as a covariate rather than an interaction) and had slightly worse fit (higher AIC and BIC) than the original model (AIC = 5477.2 versus 5472.7, BIC = 5538.5 versus 5520.8).”

This study will be of interest to psychologists and cognitive scientists interested in understanding how controllability and its subjective perception impact how people respond to stress exposure. Demonstrating that an increased sense of control buffers/protects against subsequent stress is important and may trigger further studies to characterize this phenomenon better. However, beyond the highlighted weaknesses, the current study only studied the effect of stress induction consecutive to the performance of the WS task on the same day and its generalizability is not warranted.

We thank the reviewer for this assessment and agree that we cannot assume these findings would generalise to more prolonged effects on stress responses.

**Reviewer #3 (Public review):**
Summary:This is an interesting investigation of the benefits of perceiving control and its impact on the subjective experience of stress. To assess a subjective sense of control, the authors introduce a novel wheel-stopping (WS) task where control is manipulated via size and speed to induce low and high control conditions. The authors demonstrate that the subjective sense of control is associated with experienced subjective stress and individual differences related to mental health measures. In a second experiment, they further show that an increased sense of control buffers subjective stress induced by a trier social stress manipulation, more so than a more typical stress buffering mechanism of watching neutral/calming videos.

We agree with this accurate summary of our study.

Strengths:There are several strengths to the manuscript that can be highlighted. For instance, the paper introduces a new paradigm and a clever manipulation to test an important and significant question. Additionally, it is a well-powered investigation that allows for confidence in replicability and the ability to show both high internal consistency and high external validity with an interesting set of individual difference analyses. Finally, the results are quite interesting and support prior literature while also providing a significant contribution to the field with respect to understanding the benefits of perceiving control.

We thank the reviewer for this positive assessment.

Weaknesses:There are also some questions that, if addressed, could help our readership.(1) A key manipulation was the high-intensity stressor (Anticipatory TSST signal), which was measured via subjective ratings recorded on a sliding scale at different intervals during testing. Typically, the TSST conducted in the lab is associated with increases in cortisol assessments and physiological responses (e.g., skin conductance and heart rate). The current study is limited to subjective measures of stress, given the online nature of the study. Since TSST online may also yield psychologically different results than in the lab (i.e., presumably in a comfortable environment, not facing a panel of judges), it would be helpful for the authors to briefly discuss how the subjective results compare with other examples from the literature (either online or in the lab). The question is whether the experienced stress was sufficiently stressful given that it was online and measured via subjective reports. The control condition (low intensity via reading recipes) is helpful, but the low-intensity stress does not seem to differ from baseline readings at the beginning of the experiment.

We agree that it would be helpful to expand on this further. Similar to the comment made by Reviewer 1, we wish to point out that there are ample demonstrations that such anticipatory stress inductions are effective at reliably eliciting physiological and psychological stress responses (e.g. Nasso et al., 2019; Schlatter et al., 2021; Steinbeis et al., 2015). Further, there is evidence that online versions of the TSST are also effective (DuPont et al., 2022; Meier et al., 2022), including evidence that the speech preparation phase conducted online was related to increases in heart rate and blood pressure (DuPont et al., 2022). We have elaborated further on this in our manuscript on pages 8 and 9 as follows:

“Prior research has found TSST anticipation to elicit both psychological and physiological stress responses [37-39], suggesting that the task anticipation would be a valid stress induction despite participants not performing the speech task. Moreover, prior research has validated the use of remote TSST in online settings [40, 41], including evidence that the speech preparation phase (online) was related to increased heart rate and blood pressure compared to controls [40].”

(2) The neutral videos represent an important condition to contrast with WS, but it raises two questions. First, the conditions are quite different in terms of experience, and it is interesting to consider what another more active (but not controlled per se) condition would be in comparison to the WS performance. That is, there is no instrumental action during the neutral video viewing (even passive ratings about the video), and the active demands could be an important component of the ability to mitigate stress. Second, the subjective ratings of the stress of the neutral video appear equivalent to the win condition. Would it have been useful to have a high arousal video (akin to the loss condition) to test the idea that experience of control will buffer against stress? That way, the subjective stress experience of stress would start at equivalent points after WS3.

We agree with the reviewer that this is an important issue to clarify. In our deliberations when designing this study, we considered that that any task with actionoutcome contingencies would have a degree of controllability. To better distinguish experiences of control (WS task) to an experience of no/neutral control (i.e., neither high nor low controllability), we decided to use a task in which no actions were required during the task itself. Importantly, however, there was an active demand and concentration was still required in order to perform the attention checks regarding the content of the videos and ratings of the videos.

Thank you for the suggestion of having a high arousal video condition. This would indeed be interesting to test how experiencing ‘neutral’ control and high(er) stress levels preceding the stressor task influences stress buffering and stress relief, and we have included this suggestion for future research in the discussion section (page 28) as below:

“Another avenue for future research would be to test how control buffers against stress when compared to a neutral control scenario of higher stress levels, akin to the loss domain in the WS Task, given that participants found the video condition generally relaxing. However, given that we found no differences dependent on domain for the stress induction in the WS Task conditions, it is possible that different versions of a neutral control condition would not impact the stress induction.”

(3) For the stress relief analysis, the authors included time points 2 and 3 (after the stressor and debrief) but not a baseline reading before stress. Given the potential baseline differences across conditions, can this decision be justified in the manuscript?

We thank the reviewer for raising this. Regarding the stress relief analyses (timepoints 2 and 3) and not including timepoint 1 (after the WS/video task) stress in the model, we have added to the manuscript that there was no significant difference in stress ratings between the high control and neutral control (collapsed across stress and domain) at timepoint 1 (hence why we do not think it’s necessary to include in the stress relief model). Nevertheless, we have now included a sensitivity analysis to test the Timepoint*Control interaction of stress relief when including timepoint 1 stress as a covariate. The timepoint by control interaction still holds, suggesting that the initial stress level prior to the stress induction does not impact our results of interest. The details of this analysis are included in the Sensitivity and Exploratory Analyses section on page 24:

“Although there were no significant differences between control groups in subjective stress immediately after the WS/video task (t(175.6)=1.17, p=.244), we included participants’ stress level after the WS/video task as a covariate in the stress relief analyses (Table S12). The results revealed a main effect of initial stress (β = 0.643, SE=0.040, p<.001, Table S12) on the stress relief after the stressor debrief. Compared to excluding initial stress as in the original analyses (Table 4), there was now no longer a main effect of domain (β = 0.236, SE=2.60, p=.093, Table S12), but the inference of all other effects remained the same. Importantly, there was still a significant time by control interaction (β = 9.65, SE=3.74, p=.010, Table S12) showing that the decrease in stress after the debrief was greater in the highly controllable WS condition than the neutral control video condition, even when accounting for the initial stress level.”

(4) Is the increased control experience during the losses condition more valuable in mitigating experienced stress than the win condition?

We agree that this would be helpful to clarify. To test whether the loss domain was more valuable at mitigating experiences of stress than the win condition, we ran additional analyses with just the high control condition (WS task) to test for a Domain*Time interaction. This revealed no significant Domain*Time interaction, suggesting that the stress buffering or stress relief effect was not dependent on domain in the high control conditions. These analyses are outlined in the Sensitivity and Exploratory Analyses section on page 25:

“Finally, to test whether the loss domain was more valuable at mitigating experiences of stress than the win condition, we ran additional analyses with just the high control condition (WS task) for the stress induction and stress relief to test for an interaction of domain and time. For the stress induction, there was no significant two-way interaction of domain and time (β = -1.45, SE=4.80, p=.763), nor a significant three-way interaction of domain by time by stressor intensity (β = -3.96, SE=6.74, p=.557, Table S15), suggesting that there were no differences in the stress induction dependent on domain. Similarly for the stress relief, there was no significant two-way interaction of domain and time (β = -5.92, SE=4.42, p=.182), nor a significant three-way interaction of domain by time by stressor intensity interaction (β = 8.86, SE=6.21, p=.154, Table S15), suggesting that there were no differences in the stress relief dependent on the WS Task domain.

(5) The subjective measure of control ("how in control do you feel right now") tends to follow a successful or failed attempt at the WS task. How much is the experience of control mediated by the degree of experienced success/schedule of reinforcement? Is it an assessment of control or, an evaluation of how well they are doing and/or resolution of uncertainty? An interesting paper by Cockburn et al. 2014 highlights the potential for positive prediction errors to enhance the desire for control.

We thank the reviewer for this comment. Similar to comments regarding reward rate, our task does not allow us to fully separate control from success/reinforcement because of the manipulation of difficulty. However, we did undertake sensitivity analyses and the inclusion of overall win rate accounted for limited variance when predicting stress over and above subjective control and difficulty (page 16).

“To further isolate the relationship between subjective control and stress separate from perceived task difficulty or objective task performance, we also included the overall win rate (percentage of trials won during the WS task) in the models. In Study 1, lower feelings of control were related to higher levels of subjective stress (β = -0.12, p<.001) even when controlling for both win rate (β = -0.06, p=.220) and perceived task difficulty (β = 0.37, p<.001, Table S10). This also replicated in Study 2, where lower subjective control was associated with higher feelings of stress (β = -0.32, p<.001) when controlling for perceived task difficulty (β = 0.31, p<.001) and win rate (β = -0.11, p=.428, Table S11). This suggests that there is unique variance in subjective feelings of control, separate from task performance, relevant to subjective stress.”

(6) While the authors do a very good job in their inclusion and synthesis of the relevant literature, they could also amplify some discussion in specific areas. For example, operationalizing task controllability via task difficulty is an interesting approach. It would be useful to discuss their approach (along with any others in the literature that have used it) and compare it to other typically used paradigms measuring control via presence or absence of choice, as mentioned by the authors briefly in the introduction.

We are delighted to expand on this particular point and have done so in the Discussion on page 27:

“Previous research typically accounts for different outcomes (e.g. punishment) by yoking controllable and uncontrollable conditions [3] though other work has manipulated the controllability of rewards by changing the reward rate [for example 30] where a decoy stimulus is rewarded 50% of the time in the low control condition but 80% in the high control condition. While our task design does not separate control from obtained reward, we are able to do so in the statistical analyses.”

(7) The paper is well-written. However, it would be useful to expand on Figure 1 to include (a) separate figures for study 1 (currently not included) and 2, and (b) a timeline that includes the measurements of subjective stress (incorporated in Figure 1). It would also be helpful to include Figure S4 in the manuscript.

We have expanded Figure 1 to include both Studies 1 and 2 and a timeline of when subjective stress was assessed throughout the experiment as well as adding Figure S4 to the main manuscript (now top panel within Figure 4).

**Reviewer #1 (Recommendations for the authors):**
(1) Study 2 shows a greater decrease in subjective stress after the high-control task manipulation than after the pleasant video. One possible confound is whether the amount of time to complete the WS task and the video differ. It could be helpful to look at the average completion time for the WS task and compare that to the length of the videos. Alternatively, in future studies, control for this by dynamically adjusting the video play length to each participant based on how long they took to complete the WS task.

This is an interesting suggestion. As a result, we have included the time taken as a covariate in the stress induction and stress relief analyses to ensure that any differences in time between the WS task and video task were not accounting for any of the stress induction or relief analyses. Controlling for the total time taken did not impact the stress induction or relief results. This is included in the Sensitivity and Exploratory Analyses section on page 24:

“Our second sensitivity analyses was conducted because the experiment took longer to complete for the video condition (mean = 54.3 minutes, SD = 12.4 minutes) than the WS task condition (mean = 39.7 minutes, SD = 12.8 minutes, t(186.19)=-9.32, p<.001). We therefore included the total time (in ms) as a covariate in the stress induction and stress relief analyses for Study 2. This showed that accounting for total time did not change the results of interest (Table S13), further highlighting that the time by control interactions were robust.”

(2) Because participants received feedback about their success/failure in the WS task, a confounding factor could be that they received positive feedback on highly controllable trials and negative feedback on low control trials (and/or highly difficult trials). This would suggest that it is not controllability per se that contributes to stress perception but rather feedback valence. The authors show that this is a likely factor in their results in Study 2, which shows significant effects of the loss domain on perceived control and stress. Was a similar analysis done in Study 1? Do participants receive feedback in Study 1? It would be helpful to include this information somewhere in the manuscript. I would be curious to know whether *any* feedback at all influences controllability/stress perceptions.

We thank the reviewer for this interesting suggestion. It is an interesting question as to whether feedback valence is related to stress in Study 1, and we have added this point to the Discussion on pages 27 and 28. To speak to this point, when we include the overall win rate (which captures the subsequent feedback received) when predicting subjective stress, win rate is not a significant predictor of stress over and above perceived difficulty and subjective control, suggesting that overall feedback valence may not be related to stress in Study 1. We take this as evidence that feedback may not be as important in terms of accounting for the relationship between stress and control. However, we unfortunately do not have any data in which there was no feedback provided to speak to this conclusively. This would be an interesting future study. The excerpt below is added to pages 27 and 28 of the discussion section:

“Like with perceived difficulty, we statistically accounted for reward rate and showed that the relationship between subjective control and stress was not accounted for by reward rate, for example. Similarly, participants received feedback after every trial, and thus feedback valence may contribute to stress perception. However, given that overall win rate (which captures the feedback received during the task) did not predict stress over and above perceived difficulty or subjective control, it suggests that feedback is unlikely to relate to stress over and above difficulty. Future work will need to disentangle this further to rule out such potential confounds.”

To respond specifically to the reviewer’s question about the feedback given to participants, written feedback was provided on screen to participants on a trial-bytrial basis also in Study 1 (i.e. for both studies), and we have provided more clarity about this in the manuscript on page 8 as well as providing additional details in Table S3:

“After each trial, participants were shown written feedback on screen as to whether the segment had successfully stopped on the red zone (or not), and the associated reward (or lack of). See Table S3 for details.”

(3) I'm not sure how to interpret the fact that in Figure S1, the BICs are all essentially the same. Does this mean that you don't really need all of these varying aspects of the task to achieve the same effects? Could the task be made simpler?

The similarity of BIC values suggests that a simpler WS task would have produced a worse account of the data approximately in keeping with the extent to which it is a simpler model. Here, the BIC scores for the models are similar, suggesting that adding these parameters adds explanatory power in keeping with what would have been expected from adding a parameter, but not more. We do note that the BIC is a relatively strict and conservative comparison. The fact that the most complex model overall narrowly improves parsimony; combined with the interpretable parameter values and the prior expectations given the task setup led us to focus on this most complex model.

(4) A minor point, but the authors refer to their sample as "neurotypical." Were they assessed for prior/current psychopathology/medications? If not, I might use a different term here (perhaps "non-clinical sample"), since some prior work has shown that online samples actually have higher instances of psychopathology compared to community samples.

We have changed the phrasing of ‘neurotypical’ to a ‘non-clinical sample’ as recommended.

**Reviewer #2 (Recommendations for the authors):**
Figure 4S is very informative and could be presented in the main text.

We have expanded Figure 1 to include both Studies 1 and 2 and a timeline of when subjective stress was assessed throughout the experiment as well as adding Figure S4 to the main manuscript (top panel of Figure 4).

References:

Dorfman, H. M., & Gershman, S. J. (2019). Controllability governs the balance between Pavlovian and instrumental action selection. Nature Communications, 10(1), 5826. https://doi.org/10.1038/s41467-019-13737-7

DuPont, C. M., Pressman, S. D., Reed, R. G., Manuck, S. B., Marsland, A. L., & Gianaros, P. J. (2022). An online Trier social stress paradigm to evoke affective and cardiovascular responses. Psychophysiology, 59(10), e14067. https://doi.org/10.1111/psyp.14067

Jangraw, D. C., Keren, H., Sun, H., Bedder, R. L., Rutledge, R. B., Pereira, F., Thomas, A. G., Pine, D. S., Zheng, C., Nielson, D. M., & Stringaris, A. (2023). A highly replicable decline in mood during rest and simple tasks. Nature Human Behaviour, 7(4), 596–610. https://doi.org/10.1038/s41562-023-015197

Meier, M., Haub, K., Schramm, M.-L., Hamma, M., Bentele, U. U., Dimitroff, S. J., Gärtner, R., Denk, B. F., Benz, A. B. E., Unternaehrer, E., & Pruessner, J. C. (2022). Validation of an online version of the trier social stress test in adult men and women. Psychoneuroendocrinology, 142, 105818. https://doi.org/10.1016/j.psyneuen.2022.105818

Nasso, S., Vanderhasselt, M.-A., Demeyer, I., & De Raedt, R. (2019). Autonomic regulation in response to stress: The influence of anticipatory emotion regulation strategies and trait rumination. Emotion, 19(3), 443–454. https://doi.org/10.1037/emo0000448

Schlatter, S., Schmidt, L., Lilot, M., Guillot, A., & Debarnot, U. (2021). Implementing biofeedback as a proactive coping strategy: Psychological and physiological effects on anticipatory stress. Behaviour Research and Therapy, 140, 103834. https://doi.org/10.1016/j.brat.2021.103834

Steinbeis, N., Engert, V., Linz, R., & Singer, T. (2015). The effects of stress and affiliation on social decision-making: Investigating the tend-and-befriend pattern. Psychoneuroendocrinology, 62, 138–148. https://doi.org/10.1016/j.psyneuen.2015.08.003